# Use of Airborne Radar Images and Machine Learning Algorithms to Map Soil Clay, Silt, and Sand Contents in Remote Areas under the Amazon Rainforest

Ana Carolina de S. Ferreira [1], Marcos B. Ceddia [2,3,*], Elias M. Costa [3], Érika F. M. Pinheiro [2], Mariana Melo do Nascimento [4] and Gustavo M. Vasques [5]

1 Instituto de Agronomia, Universidade Federal Rural do Rio de Janeiro, BR 465, km 7, Seropédica 23890-000, Brazil
2 Department of AgroTechnologies and Sustainability, Institute of Agronomy, Federal Rural University of Rio de Janeiro, BR 465, km 7, Seropédica 23890-000, Brazil
3 Laboratory of Water and Soils in Agroecosystem, Universidade Federal Rural do Rio de Janeiro, BR 465, km 7, Seropédica 23890-000, Brazil
4 Agronomic Engineering, Universidade Federal Rural do Rio de Janeiro, BR 465, km 7, Seropédica 23890-000, Brazil
5 Embrapa Soils, Rua Jardim Botânico 1024, Rio de Janeiro 22460-000, Brazil
* Correspondence: marcosceddia@gmail.com; Tel.: +55-21-3787-3772 or 97498-9996

**Abstract:** Soil texture has a great influence on the physical–hydric and chemical behavior of soils. In the Amazon regions, due to the presence of dense forest cover and limited access to roads, carrying out surveys and mapping of soils is challenging. When data exist, they are relatively sparse and the distribution is quite uneven. In this context, machine learning algorithms (ML) associated with remote sensor covariates offer a framework to derive digital maps of soil attributes. The objective of this study was to produce maps of surface and subsurface soil clay, silt, and sand contents in a 13.440 km$^2$ area in the Amazon. The specific objectives were to (a) evaluate the gain in prediction accuracy when using the P-band of airborne radar as a covariate; (b) evaluate two sampling approaches (Reference Area—RA and Total Area—TA); and (c) evaluate the transferability and performance of three ML algorithms: regression tree (RT), random forest (RF), and support vector machine (SVM). The study site was divided into three blocks, called Urucu, Araracanga, and Juruá, respectively. The soil dataset consisted of 151 surface and subsurface sand, silt, and clay observations and 21 covariates (20 relief variables and the backscattering coefficient from the P-band). Both the RA and TA sampling approach used 114 observations for training the prediction models (75%) and 37 for validation (25%). The RA approach was better for the development of sand and silt models. Overall, RF derived the most accurate predictions for all variables. The effect of introducing the P-band backscattering coefficient improved the sand prediction accuracy at the surface and subsurface in Araracanga, which had the highest sand content, with relative improvements (RI) of the $R^2$, root mean square error (RMSE), and mean absolute error (MAE) of 46%, 3%, and 4% at the surface, respectively, and 66.7%, 4.4%, and 5.2% at the subsurface, respectively. For silt, the P-band improved the predictions at the surface in Araracanga, which had the lowest silt contents among the blocks. For clay, adding the P-band improved the RF predictions at the subsurface, with RI of the $R^2$, RMSE, and MAE of 29%, 5%, and 5%, respectively. Despite the low observation density, inherently hindered by the low accessibility of the area and high costs of sampling thereof, the results showed the potential of ML algorithms boosted by airborne radar P-band to map soil clay, silt, and sand contents in the Amazon.

**Keywords:** digital soil mapping; soil texture; radar P-band; reference area; soil survey

## 1. Introduction

Soil texture is a fundamental physical property that strongly influences many other soil properties. The soil particle size fractions, namely clay, silt, and sand, influence soil

fertility, water infiltration and retention capacity, soil organic matter dynamics, and, thus, the ability of soils to support plants, animals and life, and secure biodiversity [1–3]. Soil sand, silt, and clay contents are input data needed for most hydrological, climatic, and environmental models. They are also used to estimate hard-to-measure soil properties such as bulk density, hydraulic conductivity, and water-holding capacity [4,5].

The Brazilian Amazon rainforest represents a major challenge for the development of systematic soil mapping studies. The region covers an immense area (59% of the Brazilian territory) and has a large portion covered by dense evergreen forest [6,7]. Additionally noteworthy is the low density of roads, with most of the territory accessed only by boat and air transport. In this region, the constant presence of clouds makes it difficult to use satellite images and aerial photos obtained by passive (optical, infrared) remote sensors [7,8]. This condition makes active sensors, such as radar, potential alternatives for observing/surveying the land, serving as support for mapping environmental patterns and resources, including soils, hydrology, geology, and geomorphology. In fact, the climatic characteristics of the Amazon region and the intense land cover by native vegetation motivated the first project of systematic mapping of the Amazon region using radar images, the RADAM Project [9], which was a pioneering effort by the Brazilian government in the 1970s to survey natural resources using airborne radar imagery. At the time, the use of side-looking airborne radar (SLAR) represented a technological advance, because the radar images could be obtained both during the day and at night and in cloudy conditions, as radar microwaves penetrate most clouds. In the RADAM project [9], the X band was used (wavelengths close to 3 cm and frequency between 8 and 12.5 GHz) and image mosaics were generated at a scale of 1:250,000. Despite the advancement in the RADAM project as a source of important maps for the Brazilian Amazon region (geological, geomorphological, soil and vegetation maps), there is still a growing demand for more detailed maps of soil attributes to support projects for different purposes, including research in soil water and carbon [7].

Among the available radar bands, for soil studies in the Amazon region under native forest, the P band is ideal because the waves can pass the clouds and the tree canopies. Most of the radar research found in the literature concentrates on forestry studies [10–16]; however, recently there has been an increase in the application of radar remote sensing for soil assessment, mainly focusing on soil moisture [17–22]. As the dielectric behavior of the soil is affected by the particle size distribution, by assessing the soil dielectric properties, radar remote sensing indirectly assesses soil particle size distribution [23]. In the Brazilian Amazon region [8], the addition of relief and vegetation covariates derived from multispectral images with distinct spatial and spectral resolutions (Landsat 8 and RapidEye) and L-band radar images (ALOS PALSAR) were evaluated for the prediction of soil organic carbon stock (CS) and particle size fractions. Overall, the results showed that, even under forest coverage, the ALOS PALSAR L-band backscattering coefficient improved the accuracy of subsurface clay content predictions (8.2% higher) from regression kriging (RK) [8].

In addition to the limited availability of P-band radar images, especially in the Amazon, the execution of soil surveys in this region faces challenges inherent to its remoteness (low accessibility, little infrastructure, high transportation costs) [7,8]. Therefore, using existing data and knowledge from soil databases and previous surveys is essential to build predictive models of attributes such as soil particle size fractions. In this sense, the Reference Area (RA) approach in association with machine learning (ML) techniques becomes strategic. The RA approach assumes that a small area, if strategically chosen, can be surveyed to build a detailed soil map or soil prediction models with the potential to be extended or applied to other (ideally larger) areas with similar soil and landscape characteristics [24,25]. In this case, the RA approach would significantly reduce mapping costs, requiring only new field studies to assess the accuracy of the predictions in the new area.

On the other hand, as soil databases are limited in remote areas, the available data do not always present density and spatial distribution of soil observations that allow the use of techniques commonly used in digital soil mapping, such as models based on multivariate statistics and geostatistics. As an alternative, machine learning (ML) algorithms have been

shown to be promising for mapping soil types and their attributes in large areas [1,4,26–31]. They refer to a large class of data-driven algorithms, some of which not following any statistical assumptions. As such, ML algorithms have the capacity of handling a large number of cross-correlated covariates (collinearity) as predictors [32].

The objectives of this study were to combine machine learning with remote sensor data to map soil surface and subsurface clay, silt, and sand fractions in the Brazilian Amazon, aiming specifically to (a) evaluate the gain in prediction accuracy from adding the P-band of airborne radar as covariate; (b) evaluate two sampling approaches (Reference Area—RA and Total Area—TA); and (c) evaluate the transferability and performance of regression tree (RT), random forest (RF), and support vector machine (SVM) models.

## 2. Materials and Methods

### 2.1. Study Area

The study area is located in the central region of the Amazonas state (at about 640 km from Manaus), covering an area of about 13.440 km$^2$ between the municipalities of Coari and Tefé (Figure 1). The area is remote and practically all covered by equatorial Amazon rainforest. The elevation ranges from 23 to 112 m above mean sea level and the climate is equatorial (Af), according to Köppen classification, with the temperature of the coldest month higher than 20 °C, mean annual precipitation of 2500 mm, and no pronounced dry period.

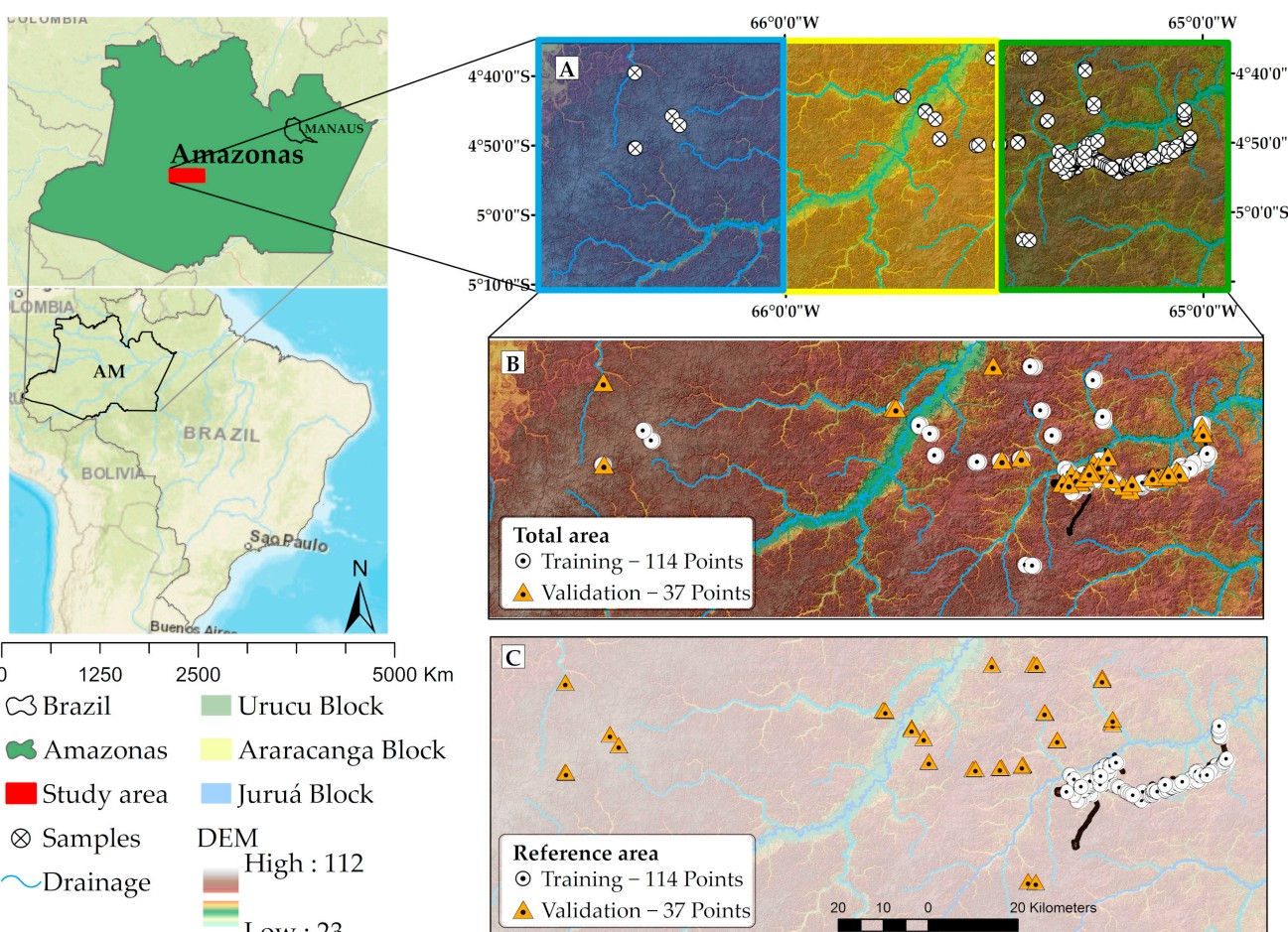

**Figure 1.** (**A**) Location of the study area in Central Amazon, Brazil; (**B**) Total area (TA) sampling, showing the 75% training and 25% validation random samples; and (**C**) Reference area (RA) sampling, with the 75% training samples concentrated at the Geólogo Pedro de Moura Support Base, and the 25% validation samples lying outside the RA.

According to ref. [8], most soils in the region have low base content, high aluminum content, and medium-to-high sand content. Some soils in the region have hydromorphic characteristics, especially those close to the floodplain of water courses and flat tops. The study area was divided into three blocks, which represent the petroleum exploration blocks by Petrobras (Brazilian Oil Company), namely Urucu (~4514 km$^2$), Araracanga (~3751 km$^2$), and Juruá (~4703 km$^2$), respectively (Figure 1A). The project database comprises data from 151 soil profiles surveyed in two field campaigns (year 2008 and 2018, respectively).

*2.2. Soil Sampling Designs*

The development of soil prediction models and maps involves financial and logistical investments to support field soil surveys and laboratory and office work. Field sampling in the Amazon is restricted by the low accessibility due to the absence of roads and limited or no infrastructure to provide essential goods and services (e.g., lodging, food, and medical services). This characteristic of the region makes the execution of soil surveys complex, especially the more detailed ones.

The Reference Area (RA) for the study was the Geólogo Pedro de Moura Support Base (BOGPM), which belongs to Petrobras (Petróleo Brasileiro S.A.) and spans across circa 80 km$^2$. The area is only accessed by air or river transport. In 2008, a detailed soil survey was carried out at BOGPM. In this area, in addition to the soil map, a database was organized containing 114 observations that included soil taxonomic class (Table 1), chemical and physical, as well as co-located relief covariates. From these data, prediction models of soil types and attributes have been developed for other areas, considering the BOGPM as an RA.

**Table 1.** Number (n) and percent of soil taxonomic classes in the 151 field observations.

| SiBCS [a] | Soil Taxonomy [b] | WRB [b] | n | Percent (%) |
|---|---|---|---|---|
| Argissolo Amarelo | *Ultisols* | *Acrisols; Lixisols* | 41 | 27.15 |
| Argissolo Vermelho | *Utisols (Typic Rhodustults)* | *Acrisols; Lixisols* | 2 | 1.32 |
| Argissolo Vermelho Amarelo | *Ultisols* | *Acrisols; Lixisols* | 29 | 19.20 |
| Argissolo Acizentado | *Ultisol (Hapludult)* | *Haplic Lixisol* | 3 | 1.98 |
| Cambissolo Háplico | *Inceptisols* | *Cambisols* | 49 | 32.45 |
| Cambissolo Flúvico | *Entisols (Fluvents)* | *Fluvisols* | 2 | 1.32 |
| Espodossolos Humilúvicos | *Spodosols (Alorthods)* | *Podzols* | 1 | 0.66 |
| Espodossolos Ferri-Humilúvicos | *Spodosols (Orthods)* | *Podzols* | 4 | 2.65 |
| Neossolo Quartzarênico | *Entisols (Quartzipsamments)* | *Arenosols* | 1 | 0.66 |
| Neossolos Flúvicos | *Entisols (Fluvents)* | *Fluvisols* | 2 | 1.32 |
| Planossolo Háplico | *Ultisols (Albaquults)* | *Planosols* | 2 | 1.32 |
| Gleissolos Háplicos | *Entisols (Aquents)* | *Gleysols; Stagnosols* | 14 | 9.27 |
| Gleissolos Melânicos | *Entisols (Fluvaquentic Humaquepts)* | *Umbric Gleysols* | 1 | 0.66 |
| Total | | | 151 | 100 |

[a] Brazilian Soil Classification System [3]. [b] Partial equivalence of the soil classes to WRB [33] and Soil Taxonomy [34].

As an RA, the BOGPM serves as a base for soil sampling, for understanding the soil–landscape relationships of the region, and for training the prediction models aiming to transfer this knowledge and derived models to a larger region expanding the soil maps and its attribute maps to remote areas at a lower cost. However, the use of the RA approach assumes that the soil and landscape data observed in the RA represent the new areas where the prediction models are intended to be applied for deriving digital maps of soils and their attributes.

In 2018, a field campaign was carried out to visit 37 new soil sites as model and map validation sites for the RA approach. In this campaign, 16 remote clearings that allowed the landing and take-off of helicopters were identified. At each clearing, soil sites located

within a 2000 m buffer were visited and sampled, expanding the original soil database from 114 to 151 soil profiles (Figure 1; Table 1).

With this data set, two sampling approaches were tested to develop soil clay, silt, and sand content prediction models for the whole area (13.440 km$^2$), which encompasses three exploration blocks (Figure 1). It is important to note that, for purposes of organizing the cartographic bases, the area was divided into exploration blocks by Petrobras. In this study, the same logic was followed for prediction and map generation. Thus, throughout this study, the names adopted for each block will be used (Urucu, Araracanga, and Juruá, as presented in Figure 1). In the first approach—Reference Area—all 114 soil profiles occurring in the RA (Figure 1C) were used for model training, while the other 37 samples outside the RA were used for external validation of the models and maps. In the second approach—Total Area—the existence of an RA was ignored inasmuch as all 151 samples were pooled together, and the 114 training (75% of the samples) and 37 validation samples (25%) were randomly drawn from the pooled database of 151 samples.

The methodological strategy to predict sand, silt and clay for each soil depth (surf and sub) is presented in the flowchart (Figure 2).

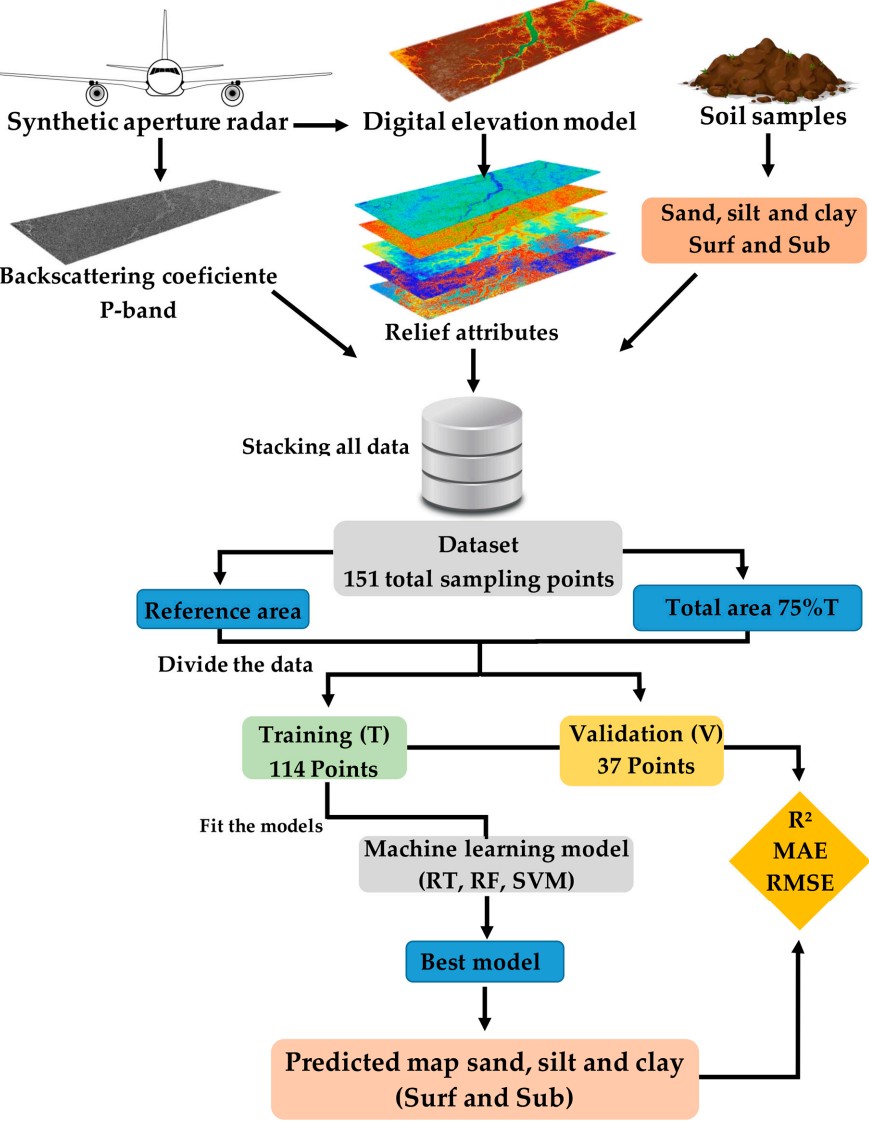

**Figure 2.** Flowchart of the methodology used for mapping soil surface (Surf) and subsurface (Sub) clay, silt, and sand contents. T—training; V—validation; RT—regression tree; RF—random forest; SVM—support vector machine; R$^2$—coefficient of determination.

### 2.3. Soil Particle Size Fractions

During the soil survey, the soil profiles were described morphologically with the separation of horizons/layers (A, AB, BA, B, C, AC, and CB, for example). For each of the horizons/layers, samples were collected for chemical and physical analyses. The sand, silt, and clay contents were determined from these samples using the Pipette method [35]. The dataset with values of sand, silt, and clay of what is called the surface layer (surf) is the weighted average of these fractions at horizons A, AB, AC, and AE (0–30 cm), while the dataset of sand, silt, and clay of the subsurface layer refers to the weighted average of these fractions in the BA, BE, and B horizons (0–100 cm) (Equation (1)). The values of the sand, silt, and clay fractions in the BC, CB, and C horizons/layers were not considered in the calculation, whereas CA and C were included when there was no B horizon, that is, for soils such as Quartzipsamments and Fluvents.

$$PSF_{surf/sub} = \sum_{i=1}^{n} PSF_i * T_i / \sum_{i=1}^{n} T_i \tag{1}$$

where: $PSF_{Surf/Sub}$ is the particle size fraction (clay, silt, or sand content) in the desired layer (surface or subsurface), in g kg$^{-1}$; $PSF_i$ is the PSF at horizon i, in g kg$^{-1}$; $T_i$ is the thickness, in m, of the portion of the horizon i that lies within the de-sired layer; and n is the number of horizons that have a portion within the desired layer.

### 2.4. Radar-Derived P-Band and Relief Covariates

The use of a radar sensor is important in the Amazon region mainly due to atmospheric conditions that include long rainy periods and the presence of clouds that often limit the use of passive remote sensors. The exclusive use of P-band (72 cm wavelength) microwave radar images in large regions covered by dense vegetation, such as the Amazon rainforest, is essential to generate thematic and relief maps. The longer wavelengths (P-band) can penetrate treetops and generate sufficiently strong reflections from the terrain below them to be more sensitive to biomass variations than other bands such as X, C, and L, and can be used to generate Digital Elevation Models (DEM).

A mosaic and a DEM of the study area were obtained from 84 Synthetic Aperture Interferometric Radar OrbiSAR-1 images, developed by Orbisat. All appropriate treatments were carried out, aiming to derive a mosaic and a DEM without interpolation failures, resulting in a hydrologically consistent DEM with 20 m spatial resolution. Primary and secondary relief derivatives were derived from the DEM using SAGA GIS version 7.7.0 [36], including Convergence Index, Topographic Wetness Index, Relative Slope Position, Channel Network Distance, Channel Network Base Level, LS-factor, Multiresolution Index of Valley Bottom Flatness, Multiresolution Index of the Ridge Top Flatness, Convexity Index, Aspect, Landforms, Profile Curvature, Plan curvature, Valley Depth, Slope Height, Mid Slope Position, Slope Gradient, Melton Ruggedness Number, and Flow Accumulation. All the data layers were brought to the same projection in ArcGIS (ESRI, Redlands, CA, USA).

The backscatter coefficient (σ°) of the HH polarization of the P-band was derived from the radar image mosaic. Reflector points in the ground were used for radiometric calibration. All calibration and radiometric corrections were performed using ENVI (L3Harris Geospatial, Broomfield, CO, USA).

### 2.5. Covariate Selection

The development of prediction models is a complex process that involves several steps. In the specific case of developing prediction models based on ML algorithms, as highlighted by ref. [32], conventionally, the choice of covariates is based on minimizing errors in input and output values. That is, a priori, no conceptual model of soil processes is contextualized. Only the processes that are transmitted by the input data are represented on the map.

In this study, two ways of covariate selection to develop ML models were tested: the "wrapper method" (WM) and "previous covariate selection" (PCS). In the first case

(WM), all the covariates were made available for the training of the ML algorithms. In the second case (PCS), two steps were followed: (a) evaluation of Pearson's correlation between particle size fractions and relief covariates, preferentially keeping the covariates with highest correlations; and (b) expert pedological knowledge was used to choose which covariates to keep on, in a case-by-case basis, aiming to better explain the soil–relief–vegetation relationships (SRV) in the region, as proposed by ref. [7].

The existence of multicollinearity was also considered both in the WM ant PCS method to make the covariates available to ML algorithms. The assessment of multicollinearity, which assesses the increase in variance due to the presence of multicollinearity, was performed based on the Variance Inflation Factor (VIF) [37], preferably keeping the covariates with VIF < 10 Equation (2).

$$\text{VIF} = 1 / \left( 1 - \text{R}^2 \right) \tag{2}$$

*2.6. Dissimilarities in Covariates between the Reference Area and Total Area*

The similarity of the landscape between the areas is important for the adequate transferability of the models. To examine the constraining effect of the relief characteristics on the transfer of the models between the reference area and the Urucu Araracanga and Juruá blocks, the descriptive statistics of the covariates were compared and the Gower similarity index (GSI; Equation (3)) [38,39] was calculated between the RA and each block, respectively.

$$\text{Sij} = \frac{1}{\text{p}} \sum\nolimits_{\text{k}=1}^{\text{p}} \left( 1 - \frac{\left| x_{\text{ik}} - x_{\text{jk}} \right|}{\text{range k}} \right) \tag{3}$$

where Sij is the GSI between sites i and j; k represents the relief variables; p is the number of variables; range k is the range of variable k.

The GSI ranges between 0 and 1. A value of 1 means maximum similarity between the sites, that is, that the sites differ in no variable, whereas 0 means that the sites differ maximally in all their variables. In the literature, the GSI is generally used in its inverted form (1—GSI), or the Gower Dissimilarity Index (GDI). In this case, the interpretation is the opposite, that is, GDI values close to 0 mean that the two sites are similar, whereas values close to 1 mean that they are dissimilar in their variables. The GDI (1—GSI) was calculated from the relief covariates plus the backscatter coefficient derived from the radar images.

*2.7. Model Training*

The soil surface and subsurface sand, clay, and silt contents were modeled by regression tree (RT) [40], random forest (RF) [41], and support vector machine (SVM) [42]. The regression tree represents a set of rules over a hierarchical sequence for the purpose of partitioning the data. Its most important feature is the ability to convert complex decision processes into a series of simple decisions [40]. The purpose of RT is to separate observations into smaller and homogeneous groups in relation to the result of interest, such as soil class or attributes [40].

Random forest consists of a large number of individual RT models trained from bootstrap samples of the data [41]. The results of all individual trees are aggregated to make a single prediction. This method can also rank the predictor variable's relative importance based on the regression prediction error of out-of-bag (OOB) predictions [41].

Support vector machine aims to determine decision limits among categories or continuous values by fitting optimal hyperplanes in the feature space that separates the samples minimizing prediction errors [42]. It can be used for classification and regression tasks. Table 2 summarizes the hyperparameters of each ML algorithms used in this study, R software environment [43].

**Table 2.** Hyperparameters of machine learning algorithms used in this study.

| Algorithms | Hyperparameters | Definition | Tuning |
|---|---|---|---|
| RT | cp | A non-negative number for complexity parameter. | 0.001–0.01 |
| | method | ANOVA | anova |
| RF | mtry | number of variables used to produce each tree | 1–10 |
| | ntree | the number of trees (default: 500) | 100–1000 |
| | nodesize | the minimum number of data points in each terminal node | 5 |
| SVM | Kernel type | the kernel function | polynomial |
| | type | svm can be used as a classification machine, as a regression machine, or for novelty detection. Depending on whether y is a factor or not, the default setting for type is C-classification or eps-regression, respectively, but may be overwritten by setting an explicit value. | 'nu-regression' or 'eps-regression' |
| | degree | parameter needed for kernel of type polynomial (default: 3) | 2–3 |
| | cost | The cost of predicting a sample within or on the wrong side of the margin. | 0–10 |
| | gamma | parameter needed for all kernels except linear (default: 1/(data dimension)) | 1 |
| | coef0 | parameter needed for kernels of type polynomial and sigmoid (default: 0) | 0 |
| | tolerance | tolerance of termination criterion (default: 0.001) | 0.001 |

RT: regression tree; RF: random forest; SVM: support vector machine.

### 2.8. Evaluation of the Accuracy of Interpolation Methods

The coefficient of determination ($R^2$; Equation (4)) was used to evaluate the goodness-of-fit of the RT, RF, and SVM models for soil sand, clay, and silt content, and the mean absolute error (MAE; Equation (5)), and the root mean square error (RMSE; Equation (6)) were used to assess their prediction accuracy.

$$R^2 = 1 - \frac{\sum_{i=1}^{n}(O_i - P_i)^2}{\sum_{i=1}^{n}(O_i - \overline{O})^2} \tag{4}$$

$$MAE = \frac{1}{n}\sum_{i=1}^{n}|O_i - P_i| \tag{5}$$

$$RMSE = \sqrt{\frac{1}{n}\sum_{i=1}^{n}(O_i - P_i)^2} \tag{6}$$

where n is the number of observations, $O_i$ and $P_i$ are the observed and predicted values, respectively, and $\overline{O}$ is the mean of observed values.

### 2.9. Evaluation of the Importance of P-Band to Model's Performance

To evaluate the importance of adding the backscattering coefficient of the P-band in the model, the Relative Improvements (RI) of the $R^2$, RMSE, and MAE were calculated, respectively (Equation (7)).

$$RI = \frac{Accuracy_{In} - Accuracy_{Out}}{Accuracy_{Out}} \times 100 \tag{7}$$

where: RI is the relative improvement, in %, accuracy is the $R^2$, MAE, or RMSE, respectively, in is the error value using the P-band, and Out is the error value without using the P-band.

The evaluation of the importance of the P-band was made for the ML models with the best performance and the covariate selection method with the best result. It was also evaluated according to the best approach (RA or TA) for each soil attribute.

## 3. Results

### 3.1. Summary Statistics

The soil sand, silt, and clay particle size fractions at the surface and subsurface layers present a frequency distribution similar to the standard normal (both skewness and excess kurtosis approximately 0), except surface clay (Table 3, whole dataset). The training and validation datasets follow the same pattern (close to normal distribution), differing in terms of minimum and maximum values, which is expected due to data partitioning. Based on the mean and median values of the particle size fractions, taken together, the textural classes vary from loan at the surface to clay loam at the subsurface. The mean and median values of sand, silt, and clay in the validation data dataset of the RA approach (V(RA)) indicate that the soils visited in remote areas outside the reference area (accessed from the 16 clearings) present the same textural classes as those observed in the reference area.

**Table 3.** Descriptive statistics of soil texture.

| Variables | Dataset | n | Min | Max | Mean | Median | SD | Sk | K | CV (%) |
|---|---|---|---|---|---|---|---|---|---|---|
| Sand Surf (g kg$^{-1}$) | W | 151 | 80 | 918 | 458 | 437 | 156 | 0.36 | −0.11 | 34 |
| | T$_{(RA)}$ | 114 | 182 | 918 | 468 | 450 | 154 | 0.48 | −0.07 | 32 |
| | V$_{(RA)}$ | 37 | 80 | 793 | 428 | 409 | 162 | 0.11 | −0.63 | 37 |
| | V$_U$ | 21 | 225 | 721 | 425 | 401 | 144 | 0.46 | −0.99 | - |
| | V$_A$ | 11 | 80 | 793 | 507 | 549 | 176 | −0.89 | 0.77 | - |
| | V$_J$ | 5 | 151 | 360 | 267 | 273 | 75 | −0.36 | −1.38 | - |
| | T$_{(TA)}$ | 114 | 80 | 883 | 451 | 435 | 150 | 0.21 | −0.36 | 33 |
| | V$_{(TA)}$ | 37 | 208 | 918 | 481 | 460 | 173 | 0.59 | −0.19 | 35 |
| Sand Sub (g kg$^{-1}$) | W | 151 | 44 | 855 | 353 | 314 | 160 | 0.50 | −0.16 | 45 |
| | T$_{(RA)}$ | 114 | 81 | 855 | 351 | 307 | 155 | 0.65 | 0.24 | 44 |
| | V$_{(RA)}$ | 37 | 44 | 695 | 357 | 338 | 178 | 0.16 | −1.09 | 49 |
| | V$_U$ | 21 | 86 | 674 | 342 | 314 | 169 | 0.41 | −1.00 | - |
| | V$_A$ | 11 | 44 | 695 | 460 | 493 | 172 | −0.97 | 0.51 | - |
| | V$_J$ | 5 | 99 | 279 | 192 | 201 | 64 | −0.12 | −1.45 | - |
| | T$_{(TA)}$ | 114 | 44 | 695 | 337 | 308 | 145 | 0.24 | −0.75 | 43 |
| | V$_{(TA)}$ | 37 | 102 | 855 | 402 | 381 | 193 | 0.54 | −0.65 | 48 |
| Silt Surf (g kg$^{-1}$) | W | 151 | 26 | 792 | 389 | 375 | 145 | 0.16 | −0.27 | 37 |
| | T$_{(RA)}$ | 114 | 26 | 687 | 364 | 351 | 131 | 0.03 | −0.12 | 36 |
| | V$_{(RA)}$ | 37 | 155 | 792 | 466 | 481 | 160 | −0.11 | −0.94 | 34 |
| | V$_U$ | 21 | 155 | 688 | 476 | 481 | 142 | −0.42 | −0.59 | - |
| | V$_A$ | 11 | 202 | 534 | 354 | 321 | 122 | 0.19 | −1.70 | - |
| | V$_J$ | 5 | 597 | 792 | 668 | 643 | 78 | 0.56 | −1.59 | - |
| | T$_{(TA)}$ | 114 | 58 | 792 | 398 | 378 | 139 | 0.21 | −0.40 | 35 |
| | V$_{(TA)}$ | 37 | 26 | 696 | 364 | 350 | 160 | 0.17 | −0.32 | 44 |
| Silt Sub (g kg$^{-1}$) | W | 151 | 84 | 600 | 339 | 340 | 105 | 0.05 | −0.21 | 31 |
| | T$_{(RA)}$ | 114 | 84 | 600 | 332 | 328 | 101 | −0.04 | 0.01 | 30 |
| | V$_{(RA)}$ | 37 | 168 | 570 | 361 | 349 | 115 | 0.14 | −1.05 | 32 |
| | V$_U$ | 21 | 191 | 570 | 359 | 343 | 113 | 0.39 | −0.87 | - |
| | V$_A$ | 11 | 168 | 486 | 309 | 303 | 104 | 0.19 | −1.42 | - |
| | V$_J$ | 5 | 388 | 551 | 480 | 479 | 61 | −0.30 | −1.61 | - |
| | T$_{(TA)}$ | 114 | 84 | 600 | 349 | 349 | 100 | 0.07 | −0.21 | 29 |
| | V$_{(TA)}$ | 37 | 112 | 582 | 309 | 306 | 116 | 0.23 | −0.43 | 37 |
| Clay Surf (g kg$^{-1}$) | W | 151 | 4 | 500 | 152 | 140 | 86 | 0.87 | 1.12 | 56 |
| | T$_{(RA)}$ | 114 | 34 | 500 | 169 | 155 | 82 | 0.79 | 1.08 | 48 |
| | V$_{(RA)}$ | 37 | 4 | 423 | 99 | 78 | 77 | 1.99 | 5.82 | 78 |
| | V$_U$ | 21 | 6 | 203 | 98 | 86 | 51 | 0.23 | −0.65 | - |
| | V$_A$ | 11 | 4 | 423 | 118 | 73 | 121 | 1.34 | 0.83 | - |
| | V$_J$ | 5 | 27 | 130 | 64 | 57 | 40 | 0.66 | −1.37 | - |
| | T$_{(TA)}$ | 114 | 4 | 500 | 152 | 139 | 90 | 0.87 | 1.10 | 59 |
| | V$_{(TA)}$ | 37 | 39 | 351 | 154 | 142 | 74 | 0.81 | 0.33 | 48 |
| Clay Sub (g kg$^{-1}$) | W | 151 | 13 | 573 | 308 | 326 | 111 | −0.28 | −0.27 | 36 |
| | T$_{(RA)}$ | 114 | 13 | 530 | 314 | 330 | 108 | −0.60 | 0.00 | 34 |
| | V$_{(RA)}$ | 37 | 70 | 573 | 288 | 267 | 120 | 0.52 | −0.43 | 42 |
| | V$_U$ | 21 | 70 | 573 | 298 | 288 | 131 | 0.36 | −0.76 | - |
| | V$_A$ | 11 | 150 | 532 | 250 | 200 | 117 | 1.12 | 0.20 | - |
| | V$_J$ | 5 | 259 | 410 | 327 | 340 | 60 | 0.13 | −1.86 | - |
| | T$_{(TA)}$ | 114 | 70 | 573 | 314 | 327 | 105 | −0.09 | −0.57 | 33 |
| | V$_{(TA)}$ | 37 | 13 | 530 | 289 | 317 | 127 | −0.49 | −0.46 | 44 |

Surf: surface; Sub: subsurface; W: whole dataset; T: training dataset; V: validation dataset; RA: reference area approach; TA: total area approach; V$_U$: Urucu block data set; V$_A$: Araracanga block data set; V$_J$: Jurua block data set n: number of observations; Min: minimum; Max: maximum; SD: standard deviation; Sk: skewness; K: kurtosis; CV: coefficient of variation.

The large coefficients of variation (CV) values (>28%) characterize the heterogeneity of sample sets in both training and validation datasets. The range of sand, silt, and clay values was high. Sand contents ranged from 80 to 918 g kg$^{-1}$ and from 44 to 855 g kg$^{-1}$ at the surface and subsurface layers, respectively.

Clay contents had similar amplitude in the two layers (4.67 to 500 on the surface and 13 to 573 on the subsurface); however, in average terms, the clay contents in the subsurface practically doubled in relation to the surface (from 152 to 308 g kg$^{-1}$). In the opposite direction, both the average levels of sand and silt tended to decrease with increasing depth (from 458 to 353 g kg$^{-1}$ for sand and from 389 to 339 g kg$^{-1}$ for silt).

The feasibility of prediction models that are based on the RA approach depends on the transferability of these models to other target areas. Thus, the statistics of the validation data of sand, silt, and clay in the three blocks (Urucu—VU, Araracanga—VA, and Juruá—VJ) separately allow a view of the similarity of the soils. The RA is located in the Urucu block, and the ideal is that the training data used there captures the great diversity of values found in all blocks. Comparing the minimum, maximum, and average sand values in the Araracanga (VA) block, both on the surface and in the subsurface, it is noted that in this region the soils had higher sand values than in the Urucu and Juruá blocks. In the first case (surface), the average sand (507 g kg$^{-1}$) was 19% higher than in the Urucu block (425 g kg$^{-1}$), while in the subsurface this difference was even greater (34%, 459 g kg$^{-1}$ in Araracanga and 342 g kg$^{-1}$ in Urucu). The statistics of silt data for the Juruá block (VJ) highlight the significant superiority of this fraction, both on the surface and in the subsurface, in relation to the other blocks. Specifically, in relation to the Urucu block (VU), the average value of silt in Juruá was 40% higher (668 g kg$^{-1}$ against 476 g kg$^{-1}$) and 34% higher (480 g kg$^{-1}$ against 359 g kg$^{-1}$), considering the surface and subsurface layers, respectively.

Additionally, in the Juruá block, the average clay content was 35% lower on the surface in relation to the data from the Urucu block (64 g kg$^{-1}$ against 98 g kg$^{-1}$). However, in the subsurface this relationship was reversed. The average clay content was 10% higher (327 g kg$^{-1}$) than that found in the Urucu block (298 g kg$^{-1}$). This inversion explains another distinction in the clay data of the Juruá block in relation to the other blocks. In Juruá, the average value of clay in the subsurface layer was 5 times higher than on the surface (64 g kg$^{-1}$ against 327 g kg$^{-1}$). In the other blocks, the increase in clay content with increasing depth was also marked but reached lower rates (3 and 2 times higher in the Urucu and Araracanga blocks, respectively).

Analyzing the statistics of sand, silt, and clay content of the validation dataset (V) using the TA approach (dataset 2), it is noted that differences of the average values in relation to training dataset (T) were lower. Only the average values of sand and silt, both at subsurface layer, presented values 10% higher than in the training dataset. In the first case (sand at subsurface) the average value was 19% higher (337 g kg$^{-1}$ against 402 g kg$^{-1}$). In the second case (silt at subsurface), the average values of the validation dataset were 11% lower than the training dataset (309 g kg$^{-1}$ against 349 g kg$^{-1}$).

Considering the evaluation of the statistics of the different granulometric fractions, in the different depths and approaches (RA and TA), it can be considered that the data present a frequency distribution close to the standard normal and that the textural classes of the soils of the reference area and the other regions visited present the same textural class (Loan and Clay loan). However, there were greater differences between the mean values of the sand, silt, and clay fractions of the validation dataset in relation to the training dataset when using the RA approach. The effect of these differences on the development and validation of prediction models is presented below, as well as the relationship between the granulometric fractions and the relief and radar covariates.

### 3.2. Similarity among the Reference Area and Exploration Blocks

Table 4 presents the statistics of the prediction covariates. Comparing the data between the blocks, it is noted that the region of the Juruá block was the one with the greatest

discrepancy in relation to the reference area. Some relief covariates in the Juruá block had very different minimum, maximum, average, and median values compared with the RA, which reinforce the dissimilarity between these landscapes (Table 4). The covariates CNBL, CND, MRRTF, and MRVBF stand out as those with the most different relief statistics in the Juruá region in relation to RA (Table 4).

**Table 4.** Descriptive statistics of the covariates in the study area by blocks.

| Covariates (Unity) | Reference Area (199,167 Pixels) | | | | | Urucu (11,209,198 Pixels) | | | | |
|---|---|---|---|---|---|---|---|---|---|---|
| | Mean | Median | SD | Min | Max | Mean | Median | SD | Min | Max |
| CI (d) | 0.03 | 0.59 | 16.80 | −94.51 | 96.07 | −0.0002 | 0.54 | 16.41 | −98.08 | 98.91 |
| TWI (d) | 7.66 | 7.56 | 1.06 | 4.61 | 12.30 | 8.07 | 7.98 | 1.23 | 4.33 | 12.54 |
| RSP (0–1) | 0.48 | 0.51 | 0.30 | 0 | 1 | 0.44 | 0.45 | 0.30 | 0 | 1 |
| CND (m) | 6.40 | 6.15 | 4.01 | 0 | 25.39 | 5.41 | 4.88 | 3.95 | 0 | 29.64 |
| CNBL (m) | 61.72 | 61.16 | 5.95 | 46.56 | 79.59 | 63.47 | 64.07 | 7.16 | 23.03 | 83.16 |
| MRVBF (d) | 5.73 | 9.38 | 4.52 | 0 | 9.98 | 6.69 | 9.82 | 4.33 | 0 | 9.98 |
| MRRFT (d) | 2.84 | 1.97 | 2.67 | 0 | 7.93 | 4.02 | 4.76 | 3.09 | 0 | 7.99 |
| CXI (d) | 51.34 | 52.41 | 7.63 | 0.15 | 69.19 | 50.29 | 51.85 | 8.89 | 0 | 73.19 |
| ASP (°) | 177.10 | 175.22 | 106.81 | 0 | 360 | 173.78 | 171.04 | 107.03 | 0 | 360 |
| LF (d) | 5.32 | 5.00 | 2.41 | 1.00 | 10.00 | 5.18 | 5.00 | 2.11 | 1.00 | 10.00 |
| ProfC (m⁻¹) | −0 | −0 | 0 | −0.009 | 0.01 | −0 | 0 | 0 | −0.013 | 0.011 |
| PlanC (m⁻¹) | 0.0 | 3.40 | 0.0 | −0.007 | 0.01 | 0 | 0 | 0 | −0.010 | 0.013 |
| SH (m) | 4.08 | 3.55 | 1.85 | 1.47 | 18.94 | 3.84 | 3.36 | 1.79 | 1.13 | 25.51 |
| MSP (%) | 0.27 | 0.25 | 0.17 | 0.00 | 0.82 | 0.25 | 0.23 | 0.16 | 0.00 | 0.85 |
| S (%) | 6.23 | 5.15 | 4.87 | 0.00 | 48.86 | 5.16 | 3.70 | 4.77 | 0.00 | 67.20 |
| MR (d) | 0.25 | 0.16 | 0.29 | 0.00 | 2.49 | 0.21 | 0.10 | 0.27 | 0.00 | 2.95 |
| FC (d) | 2451 | 2996 | 3090 | 400 | 81207 | 2347 | 1449 | 2956 | 400 | 14170 |
| P-band (σ°) | 0.43 | 0.43 | 0.07 | 0 | 0.99 | 0.44 | 0.44 | 0.06 | 0 | 0.90 |

| Covariates (Unity) | Araracanga (9,364,993 Pixels) | | | | | Juruá (11,730,902 Pixels) | | | | |
|---|---|---|---|---|---|---|---|---|---|---|
| | Mean | Median | SD | Min | Max | Mean | Median | SD | Min | Max |
| CI (d) | 0 | 0.49 | 16.45 | −98.78 | 99.01 | 0.00 | 0.78 | 18.10 | −99.21 | 99.40 |
| TWI (d) | 7.92 | 7.72 | 1.41 | 4.36 | 12.37 | 7.58 | 7.38 | 1.28 | 3.86 | 12.01 |
| RSP (0–1) | 0.41 | 0.41 | 0.31 | 0 | 1 | 0.35 | 0.32 | 0.29 | 0 | 1 |
| CND (m) | 6.01 | 5.32 | 4.83 | 0 | 33.92 | 4.45 | 3.42 | 4.08 | 0 | 40.50 |
| CNBL (m) | 63.93 | 65.28 | 8.85 | 34.16 | 85.97 | 76.03 | 77.62 | 8.40 | 49.88 | 95.63 |
| MRVBF (d) | 4.96 | 4.77 | 4.13 | 0 | 9.96 | 3.70 | 3.89 | 2.82 | 0 | 9.65 |
| MRRFT (d) | 3.37 | 2.67 | 3.15 | 0 | 9.73 | 6.53 | 9.36 | 4.19 | 0 | 9.98 |
| CXI (d) | 48.32 | 50.92 | 11.07 | 0 | 73.40 | 39.58 | 41.13 | 8.27 | 0 | 63.48 |
| ASP (°) | 171.04 | 168.26 | 109.06 | 0 | 360 | 168.08 | 166.38 | 109.74 | 0 | 360 |
| LF (d) | 5.26 | 5.00 | 2.32 | 1.00 | 10.00 | 5.32 | 5.00 | 2.03 | 1.00 | 10.00 |
| ProfC (m⁻¹) | −0.0 | 0.0 | 0.0 | −0.011 | 0.012 | −0.0 | −0.0 | 0 | −0.014 | 0.016 |
| PlanC (m⁻¹) | 0.0 | 0.0 | 0 | −0.012 | 0.011 | 0.0 | 0.0 | 0 | −0.013 | 0.018 |
| SH (m) | 4.18 | 3.59 | 2.11 | 1.16 | 27.33 | 3.62 | 3.12 | 1.73 | 1.14 | 32 |
| MSP (%) | 0.31 | 0.29 | 0.20 | 0 | 0.88 | 0.22 | 0.18 | 0.16 | 0 | 0.89 |
| S (%) | 5.81 | 4.25 | 5.34 | 0 | 50.21 | 5.39 | 4.02 | 5.24 | 0 | 76.92 |
| MR (d) | 0.24 | 0.11 | 0.32 | 0 | 3.01 | 0.18 | 0.00 | 0.27 | 0 | 4.23 |
| FC (d) | 2332 | 1421 | 2993 | 400 | 13304 | 1609 | 1059 | 1735 | 400 | 6948 |
| P-band (σ°) | 0.45 | 0.45 | 0.11 | 0 | 0.93 | 0.43 | 0.43 | 0.10 | 0 | 0.94 |

RA: Referencea area; U: block Urucu; A: block Araracanga; J: block Jurua; n: number of observations; Referencea area: (n = 199,167); Urucu: (n = 11,209,198); n Araracanga: (n = 9,364,993); n Jurua: (n = 11,730,902); Min: minimum; Max: maximum; SD: standard deviation. d: dimensionless. CI—Convergence Index; TWI—Topographic Wetness Index; RSP—Relative Slope Position; CND—Channel Network Distance; CNBL—Channel Network Base Level;; MRVBF—Multiresolution Index of Valley Bottom Flatness; MRRFT—Multiresolution Index of the Ridge Top Flatness; CXI—Convexity Index; ASP—Aspect; LF—Landforms; ProfC—Profile Curvature; PlanC—Plan curvature; SH—Slope Height; MSP—Mid Slope Position; S—Slope Gradient; MR—Melton Ruggedness; FC—Flow Accumulation; P-band.

In Figure 3A–C, graphs are presented with the general GDI (red bars) and the same index for each covariate (gray bars). According to the GDI values, the RA and the Urucu, Araracanga, and Juruá blocks were similar in their relief variables, with GDI values of 0.155, 0.164, and 0.171, respectively (Figure 3). The dissimilarity increased by about 10% departing from the Urucu block towards Juruá (farthest from the reference area). The areas with the highest GDI were those associated with lowland areas (hydromorphic lowlands—black

arrows on maps) and higher regions located at watershed upper boundaries (pixels with more discrete values highlighted with blue arrows on maps). The relief covariates that contributed most to differentiate the blocks in relation to RA were MRVBF, MRRTF, and RSP. These covariates were also the ones that had the highest correlations with the soil particle size fractions under study (Figure 4). From the results seen in Figure 3, the GDI can be used to both support the choice or to change a previously selected RA. In this study, the RA was imposed because it is the only accessible area in the region. However, it is possible to conjecture that if we were to change the RA, this change should be in the sense of including regions that expand the expression of the covariates that most differentiated the exploration blocks in relation to the RA (in this case, MRVBF, MRRTF, and RSP). It is important to highlight that these areas are the most difficult to access and cause the most undersampling in these environments.

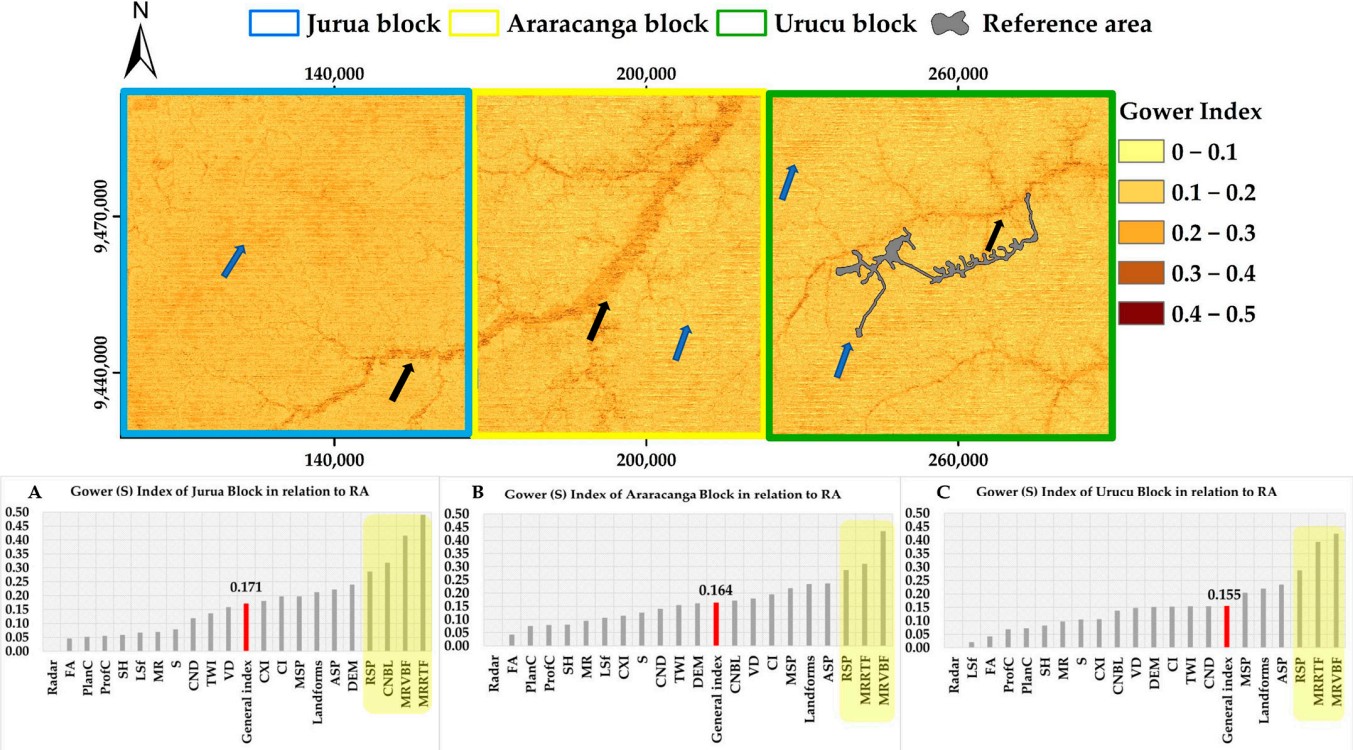

**Figure 3.** Gower index by covariate (blue bars) and general Gower index (red bar and corresponding value) among the reference area and Juruá, Araracanga and Urucu (**A–C**, respectively). The covariates that contributed with the greatest dissimilarity (GDI > 0.25) are highlighted in yellow. DEM—Digital Elevation Model; CI—Convergence Index; TWI—Topographic Wetness Index; RSP—Relative Slope Position; CND—Channel Network Distance; CNBL—Channel Network Base Level; LFf—LS-factor; MRVBF—Multiresolution Index of Valley Bot-tom Flatness; MRRFT—Multiresolution Index of the Ridge Top Flatness; CXI—Convexity Index; ASP—Aspect; LF—Landforms; ProfC—Profile Curvature; PlanC—Plan curvature; VD—Valley Depth; SH—Slope Height; MSP—Mid Slope Position; S—Slope Gradient; MR—Melton Ruggedness; FC—Flow Accumulation; P-band.

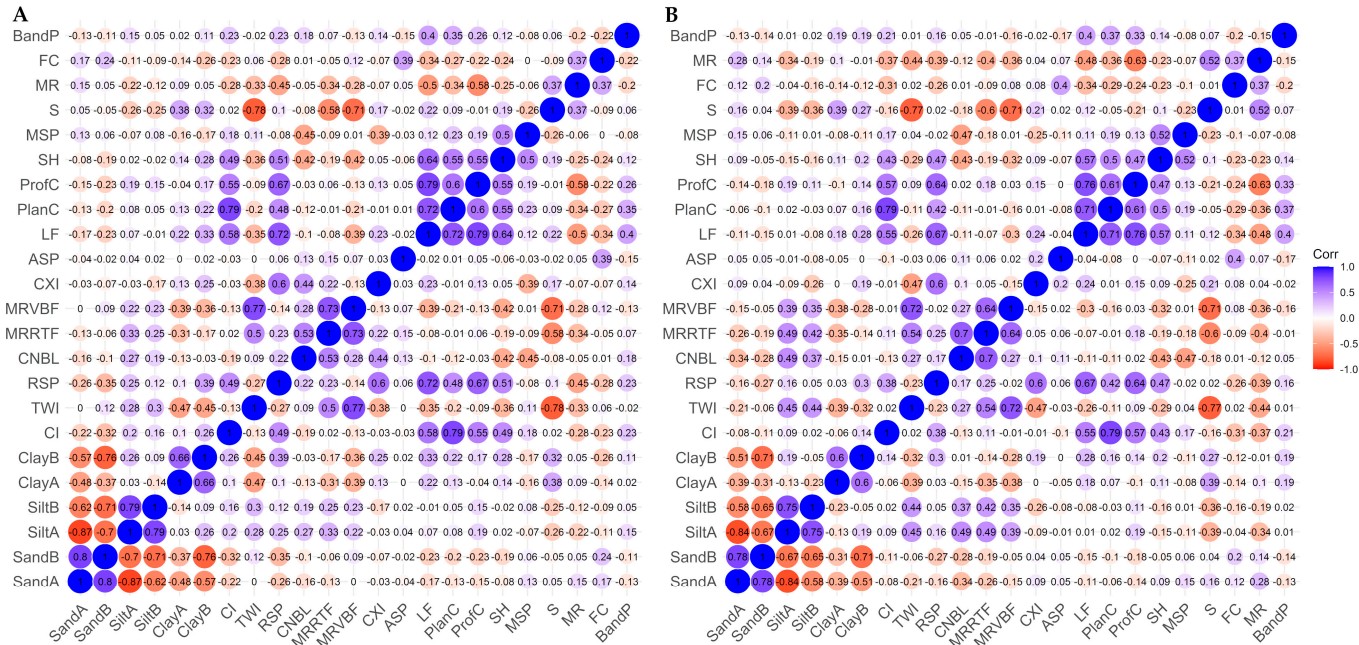

**Figure 4.** Correlation matrices of remote sensing covariates against soil particle size fractions in the training datasets from the reference area (**A**) and total area (**B**). ClayA, SiltA, SandA: surface soil particle sizes fractions; ClayB, SiltB, SandB: subsurface soil particle sizes fractions. A strong blue circle has a maximum positive correlation, a strong red circle has a maximum negative correlation. Between these two, the colour tone decreases as the correlation decreases. CI—Convergence Index; TWI—Topographic Wetness Index; RSP—Relative Slope Position; CND—Channel Network Distance; CNBL—Channel Network Base Level; MRVBF—Multiresolution Index of Valley Bottom Flatness; MRRFT—Multiresolution Index of the Ridge Top Flatness; CXI—Convexity Index; ASP—Aspect; LF—Landforms; ProfC—Profile Curvature; PlanC—Plan curvature; SH—Slope Height; MSP—Mid Slope Position; S—Slope Gradient; MR—Melton Ruggedness; FC—Flow Accumulation; P-band.

From the data obtained (Table 4 and Figure 3), it appears that although there were differences in the statistics of the covariates of the blocks in relation to the reference area, the Gower index of similarity showed that the blocks had a very low dissimilarity value, indicating that the models developed in the reference area have the potential to be transferred to other areas.

### 3.3. Remote Sensing Covariates and Soil Particle Size Fractions Relationships

In both the RA and TA training datasets, all covariates had correlations lower than 0.50 against soil particle size fractions (Figure 4). In the RA dataset (Figure 4A), the highest correlation values for each particle size fraction were found between the topographic wetness index (TWI) and clay at the surface (−0.47) and subsurface (−0.45), surface silt (0.33) and multiresolution index of ridge top flatness (MRRTF), subsurface silt and TWI (0.30), and relative slope position (RSP) and sand at the surface (−0.26) and subsurface (−0.35). In the TA dataset (Figure 4B), the highest correlations were surface clay against slope (0.39) or TWI (−0.39), subsurface clay against TWI (−0.32), surface silt against channel network base level (CNBL) (0.49) or MRRTF (0.49), subsurface silt against TWI (0.44), surface sand against CNBL (−0.34), and subsurface sand against CNBL (−0.28). Overall, sand content had the lowest correlations against remote sensing covariates.

The results of the general Gower index (Figure 3) showed that there was little dissimilarity between the RA and the Urucu, Araracanga, and Juruá blocks, with GDI (values of 0.155, 0.164, and 0.171, respectively). However, even though these dissimilarity values are low, most of the covariates that had higher correlations (Figure 4) also had greater

contributions of dissimilarity index values in relation to the general Gower index (RSP, CI, MRVBF, MRRTF, LF) (Figure 3).

Importance of predictor covariates for the attributes evaluated in the RF model is seen in (Figure 5).

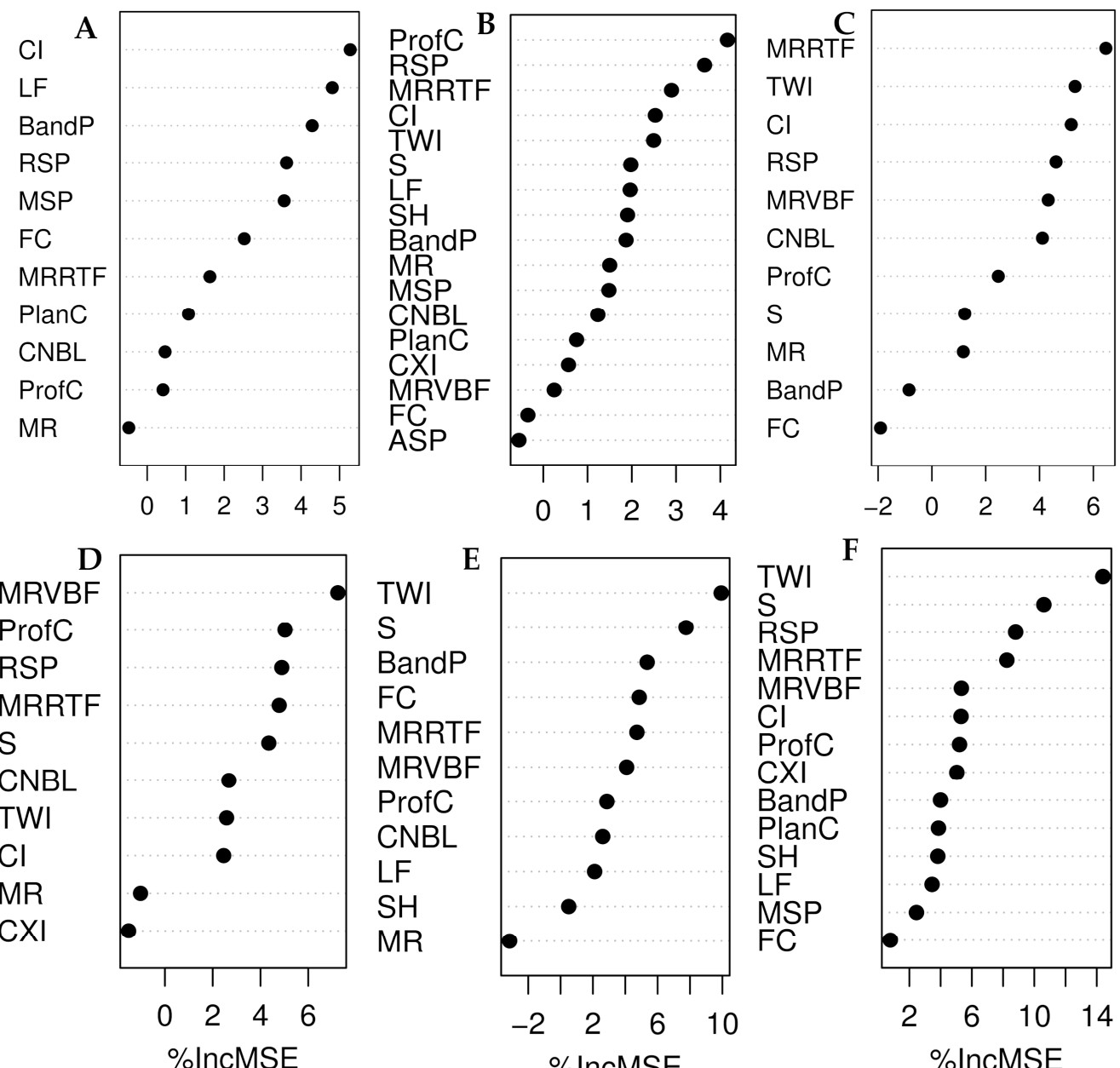

**Figure 5.** Importance of predictor covariates for the attributes evaluated in the RF model. (**A**) Sand Surf (**B**); Sand Sub; (**C**) Silt Surf; (**D**) Silt Sub; (**E**) Clay Surf; (**F**) Clay Sub. Surf—surface; Sub—subsurface.

The source material, relief, vegetation, and climate act in tandem to explain the spatial distribution patterns of soil types in the region. These same covariates were contextualized in the soil–relief–vegetation model (SRV) (Figure 6).

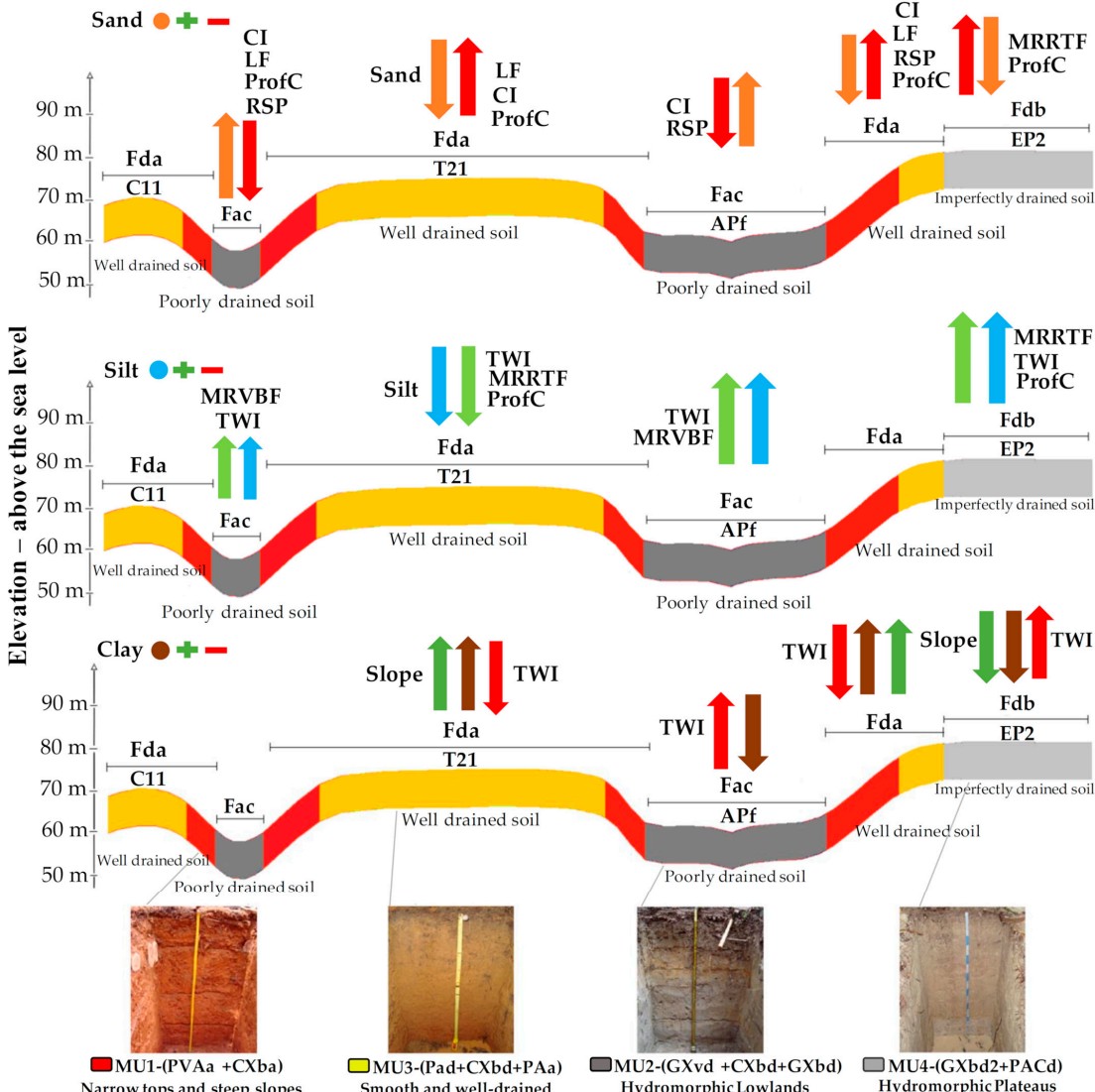

**Figure 6.** Soil–relief–vegetation relationships for soil sand, silt, and clay contents in the study area. Green arrow—positive correlation with the covariate; red arrow—negative correlation with the covariate. MU—mapping unit; Fac—Flooded Plain Open Tropical Forest; FDA—Dense Highland Tropical Forest; Fdb—Planalto Open Tropical Forest; APf—River plains; C11—Well-drained flat top areas; T21—Tabular Interfluves; EP2—Biplain-plain surfaces; H.S.—Holocene Sediments; P.S.—Pleistocene Sediments. (Source: modified from ref. [7]).

In general, the covariates convergence index (CI), landforms (LF), radar P-band backscattering coefficient (P-band), profile curvature (ProfC), RSP, and MRRTF were the most important for sand prediction by RF (Figure 5A,B). The CI represents the behavior of the surface runoff, which was influenced by the shapes of the terrain, represented by LF. The sand contents were higher close to river channels, where CI values were negative, meaning converging terrains towards lowland channels. Positive CI values indicate divergent areas, where well-drained tops and flatter slopes predominate, from which surface runoff occurs in all directions. In these areas the sand contents were lower. The RSP was applied to identify topographical features and its values ranged from 0 to 1. The values closer to 0 were characterized by lowland regions, that is, the V- and U-shaped valleys, which have high levels of sand. Values closer to 1 represent upper slopes and ridge tops with low sand contents. The profile curvature (ProfC) expressed the difference between convex curvatures of the concave ones, influencing the surface flow velocity from the higher to the lower parts (Figure 6). It also allowed greater distinction between well-drained soils on ridge tops

(convex surfaces) and imperfectly drained soils on concave to flat surfaces, for instance in V- and U-shaped valleys.

The covariates MRRTF, TWI, multiresolution index of valley bottom flatness (MRVBF), and ProfC had positive correlations with silt. The flat tops on the uplands were represented by high MRRTF values, whereas the valley bottoms had the highest MRVBF values. These covariates, associated with TWI, characterize the spatial distribution of soil saturation zones, adding important information to locate hydromorphic soils. In turn, these zones had the higher silt contents (Figure 6) and are the zones where the lowlands (MU2—Aquents, Aquepts) and uplands with flat tops (MU4—Aquults, Aquents) occur. Again, the ProfC helped to separate the areas of well-drained soils (convex surfaces) from those with imperfectly drained ones (concave to flat surfaces), mainly at the subsurface.

The combination of the slope and TWI covariates allowed identifying the regions with the highest clay contents, where the MU1 (Ultisols, Inceptisols) and MU3 (Ultisols, Inceptisols) units are found (Figure 6). The MU1 regions were represented by steeper slopes generally closer to large drainage networks where the slope influences the speed of surface and subsurface flows. The slope has great potential to help in the identification of Ultisols areas where the highest clay contents predominate. The MU3 unit occurs on well-drained tops with smoother slopes and relatively flat to smoothly wavy relief with good drainage, also with high clay contents (Figure 6).

### 3.4. Model Prediction Performance

Random forest derived the best predictions, with the least errors, for all soil particle size fractions at both layers, followed by SVM (Tables 5–8). Regression tree is the simplest among the three methods tested. It creates a series of decision rules based on the covariates to make a prediction at a terminal leaf. As such, it was uncapable of outperforming RF, which is a combination of RTs, and SVM. On the other hand, RF outperformed SVM, meaning that decision rules derived from a series of RTs are better than a single set of hyperplanes. In fact, in general the prediction errors were more similar between RT and SVM than between SVM and RF. Favoring RF is the fact that it uses random selections of covariates and training and validation (OOB) sets for building each tree, which control overfitting minimizing validation errors.

**Table 5.** Accuracy assessment soil surface and subsurface sand content predictions using the Reference Area (RA) sampling design.

| | | RT | | | RF | | | SVM | | |
|---|---|---|---|---|---|---|---|---|---|---|
| **Atributtes** | **Data** | $R^2$ | **RMSE** | **MAE** | $R^2$ | **RMSE** | **MAE** | $R^2$ | **RMSE** | **MAE** |
| Sand Surf PCS | T | 0.34 | 124 | 96 | 0.93 | 67 | 53 | 0.47 | 113 | 91 |
| | $V_U$ | 0.09 | 144 | 117 | 0.24 | 131 | 113 | 0.07 | 141 | 125 |
| | $V_{UA}$ | 0.03 | 165 | 129 | 0.19 | 140 | 116 | 0.01 | 162 | 140 |
| | V | 0.01 | 176 | 135 | 0.24 | 144 | 120 | 0.08 | 173 | 149 |
| | $V_{UJ}$ | 0.03 | 166 | 128 | 0.31 | 138 | 119 | 0.12 | 162 | 141 |
| Sand Surf WM | T | 0.36 | 122 | 96 | 0.94 | 67 | 53 | 0.57 | 106 | 86 |
| | $V_U$ | 0.21 | 132 | 103 | 0.20 | 132 | 114 | 0.04 | 144 | 129 |
| | $V_{UA}$ | 0.06 | 164 | 128 | 0.18 | 141 | 116 | 0.20 | 140 | 118 |
| | V | 0.03 | 175 | 132 | 0.19 | 148 | 123 | 0.19 | 145 | 123 |
| | $V_{UJ}$ | 0.09 | 157 | 113 | 0.22 | 143 | 125 | 0.08 | 151 | 134 |
| Sand Sub PCS | T | 0.45 | 114 | 90 | 0.92 | 63 | 50 | 0.47 | 113 | 95 |
| | $V_U$ | 0.09 | 161 | 137 | 0.24 | 147 | 126 | 0.20 | 148 | 126 |
| | $V_{UA}$ | 0.01 | 181 | 152 | 0.15 | 163 | 140 | 0.18 | 166 | 141 |
| | V | 0.00 | 190 | 155 | 0.11 | 165 | 143 | 0.24 | 168 | 139 |
| | $V_{UJ}$ | 0.02 | 179 | 145 | 0.17 | 154 | 132 | 0.21 | 155 | 127 |
| Sand Sub WM | T | 0.48 | 111 | 87 | 0.92 | 64 | 51 | 0.57 | 105 | 86 |
| | $V_U$ | 0.14 | 159 | 133 | 0.36 | 137 | 113 | 0.13 | 154 | 128 |
| | $V_{UA}$ | 0.05 | 181 | 155 | 0.25 | 158 | 134 | 0.15 | 173 | 146 |
| | V | 0.00 | 194 | 163 | 0.16 | 162 | 138 | 0.17 | 167 | 141 |
| | $V_{UJ}$ | 0.03 | 183 | 149 | 0.25 | 147 | 123 | 0.17 | 148 | 124 |

PCS—Previous Covariate Selection; WM—Wrapper Method; T: Training dataset; V: validation data set; $V_U$: Urucu block validation dataset; $V_{UA}$: Urucu/Araracanga block validation dataset; V: Urucu/Araracanga/Jurua block validation dataset; $V_{UJ}$: Urucu/Jurua block validation dataset.

**Table 6.** Accuracy assessments of soil surface and subsurface silt content predictions using the Reference Area (RA).

| Atributtes | Data | RT $R^2$ | RMSE | MAE | RF $R^2$ | RMSE | MAE | SVM $R^2$ | RMSE | MAE |
|---|---|---|---|---|---|---|---|---|---|---|
| Silt Surf PCS | T | 0.49 | 93 | 72 | 0.91 | 56 | 43 | 0.50 | 93 | 71 |
| | $V_U$ | 0.19 | 163 | 138 | 0.58 | 130 | 112 | 0.33 | 130 | 107 |
| | $V_{UA}$ | 0.07 | 154 | 129 | 0.37 | 120 | 99 | 0.17 | 175 | 123 |
| | V | 0.07 | 175 | 144 | 0.36 | 141 | 114 | 0.28 | 185 | 133 |
| | $V_{UJ}$ | 0.18 | 189 | 158 | 0.52 | 155 | 131 | 0.38 | 156 | 124 |
| Silt Surf WM | T | 0.46 | 95 | 73 | 0.92 | 55 | 42. | 0.58 | 87 | 65 |
| | $V_U$ | 0.26 | 163 | 144 | 0.46 | 139 | 120 | 0.24 | 143 | 122 |
| | $V_{UA}$ | 0.06 | 157 | 136 | 0.26 | 128 | 106 | 0.13 | 149 | 119 |
| | V | 0.08 | 174 | 149 | 0.26 | 149 | 122 | 0.22 | 159 | 128 |
| | $V_{UJ}$ | 0.26 | 186 | 161 | 0.42 | 164 | 140 | 0.26 | 158 | 134 |
| Silt Sub PCS | T | 0.47 | 73 | 58 | 0.91 | 43 | 32 | 0.39 | 79 | 61 |
| | $V_U$ | 0.36 | 90 | 72 | 0.51 | 89 | 71 | 0.38 | 91 | 77 |
| | $V_{UA}$ | 0.38 | 86 | 72 | 0.41 | 88 | 73 | 0.33 | 111 | 91 |
| | V | 0.26 | 99 | 80 | 0.46 | 89 | 74 | 0.39 | 131 | 101 |
| | $V_{UJ}$ | 0.22 | 106 | 83 | 0.56 | 90 | 73 | 0.39 | 126 | 93 |
| Silt Sub WM | T | 0.49 | 72 | 57 | 0.92 | 43 | 32 | 0.53 | 72 | 56 |
| | $V_U$ | 0.35 | 89 | 72 | 0.42 | 93 | 74 | 0.42 | 84 | 67 |
| | $V_{UA}$ | 0.33 | 89 | 73 | 0.31 | 93 | 76 | 0.39 | 91 | 76 |
| | V | 0.22 | 102 | 81 | 0.37 | 94 | 78 | 0.39 | 115 | 89 |
| | $V_{UJ}$ | 0.21 | 106 | 83 | 0.50 | 95 | 78 | 0.37 | 120 | 88 |

PCS—Previous Covariate Selection; WM—Wrapper Method; T: Training dataset; V: validation data set; $V_U$: Urucu block validation dataset; $V_{UA}$: Urucu/Araracanga block validation dataset; V: Urucu/Araracanga/Jurua block validation dataset; $V_{UJ}$: Urucu/Jurua block validation dataset.

**Table 7.** Accuracy assessment of soil surface and subsurface clay content predictions using the Reference Area (RA) sampling design.

| Atributtes | DATA | RT $R^2$ | RMSE | MAE | RF $R^2$ | RMSE | MAE | SVM $R^2$ | RMSE | MAE |
|---|---|---|---|---|---|---|---|---|---|---|
| Clay Surf PCS | T | 0.53 | 55 | 41 | 0.91 | 31 | 23 | 0.47 | 61 | 45 |
| | $V_U$ | 0.09 | 73 | 59 | 0.24 | 71 | 59 | 0.21 | 67 | 53 |
| | $V_{UA}$ | 0.04 | 90 | 70 | 0.02 | 92 | 72 | 0.08 | 115 | 73 |
| | V | 0.03 | 90 | 73 | 0.02 | 92 | 73 | 0.04 | 111 | 73 |
| | $V_{UJ}$ | 0.06 | 78 | 65 | 0.19 | 76 | 64 | 0.17 | 69 | 57 |
| Clay Surf WM | T | 0.54 | 54 | 40 | 0.92 | 31 | 23 | 0.56 | 56 | 41 |
| | $V_U$ | 0.08 | 74 | 59 | 0.18 | 71 | 59 | 0.27 | 65 | 50 |
| | $V_{UA}$ | 0.04 | 89 | 70 | 0.02 | 91 | 71 | 0.17 | 82 | 61 |
| | V | 0.03 | 90 | 73 | 0.02 | 91 | 72 | 0.10 | 96 | 72 |
| | $V_{UJ}$ | 0.05 | 78 | 66 | 0.15 | 75 | 63 | 0.15 | 91 | 68 |
| Clay Sub PCS | T | 0.61 | 67 | 53 | 0.91 | 39 | 30 | 0.58 | 70 | 52 |
| | $V_U$ | 0.16 | 119 | 90 | 0.20 | 114 | 86 | 0.14 | 120 | 93 |
| | $V_{UA}$ | 0.02 | 136 | 101 | 0.08 | 122 | 95 | 0.17 | 117 | 95 |
| | V | 0.02 | 130 | 93 | 0.07 | 116 | 89 | 0.13 | 113 | 92 |
| | $V_{UJ}$ | 0.15 | 113 | 81 | 0.18 | 107 | 80 | 0.12 | 114 | 90 |
| Clay Sub WM | T | 0.62 | 65 | 52 | 0.92 | 38 | 29 | 0.65 | 65 | 49 |
| | $V_U$ | 0.02 | 138 | 103 | 0.18 | 115 | 87 | 0.07 | 128 | 99 |
| | $V_{UA}$ | 0.00 | 146 | 111 | 0.08 | 120 | 93 | 0.14 | 118 | 93 |
| | V | 0.00 | 141 | 104 | 0.07 | 114 | 88 | 0.03 | 152 | 116 |
| | $V_{UJ}$ | 0.03 | 131 | 95 | 0.17 | 108 | 81 | 0.02 | 170 | 131 |

PCS—Previous Covariate Selection; WM—Wrapper Method; T: Training dataset; V: validation data set; $V_U$: Urucu block validation dataset; $V_{UA}$: Urucu/Araracanga block validation dataset; V: Urucu/Araracanga/Jurua block validation dataset; $V_{UJ}$: Urucu/Jurua block validation dataset.

**Table 8.** Accuracy assessment of soil surface and subsurface sand, silt, and clay content predictions using the Total Area (TA) sampling design.

| | | RT | | | RF | | | SVM | | |
|---|---|---|---|---|---|---|---|---|---|---|
| Atributtes | Data | $R^2$ | RMSE | MAE | $R^2$ | RMSE | MAE | $R^2$ | RMSE | MAE |
| Sand Surf PCS | T114 | 0.51 | 104 | 79 | 0.93 | 62 | 49 | 0.52 | 105 | 84 |
| | V37 | 0.00 | 198 | 152 | 0.11 | 161 | 124 | 0.15 | 163 | 127 |
| Sand Surf WM | T114 | 0.51 | 104 | 79 | 0.94 | 64 | 50 | 0.77 | 73 | 44 |
| | V37 | 0.00 | 198 | 152 | 0.13 | 159 | 124 | 0.03 | 209 | 158 |
| Sand Sub PCS | T114 | 0.54 | 97 | 80 | 0.93 | 58 | 47 | 0.40 | 113 | 94 |
| | V37 | 0.03 | 202 | 148 | 0.23 | 174 | 137 | 0.21 | 180 | 138 |
| Sand Sub WM | T114 | 0.55 | 97 | 80 | 0.95 | 59 | 48 | 0.81 | 64 | 41 |
| | V37 | 0.03 | 202 | 148 | 0.22 | 177 | 145 | 0.19 | 207 | 162 |
| Silt Surf PCS | T114 | 0.58 | 89 | 69 | 0.91 | 53 | 41 | 0.50 | 98 | 78 |
| | V37 | 0.04 | 182 | 140 | 0.14 | 147 | 113 | 0.20 | 142 | 108 |
| Silt Surf WM | T114 | 0.92 | 54 | 42 | 0.92 | 54 | 42 | 0.60 | 89 | 72 |
| | V37 | 0.17 | 144 | 111 | 0.17 | 144 | 111 | 0.14 | 147 | 112 |
| Silt Sub PCS | T114 | 0.49 | 71 | 57 | 0.91 | 39 | 31 | 0.42 | 76 | 61 |
| | V37 | 0.06 | 123 | 97 | 0.03 | 120 | 98 | 0.29 | 102 | 79 |
| Silt Sub WM | T114 | 0.51 | 69 | 55 | 0.92 | 38 | 30 | 0.55 | 69 | 54 |
| | V37 | 0.04 | 126 | 99 | 0.06 | 116 | 94 | 0.21 | 107 | 84 |
| Clay Surf PCS | T114 | 0.56 | 58 | 44 | 0.91 | 34 | 25 | 0.59 | 60 | 46 |
| | V37 | 0.23 | 71 | 58 | 0.23 | 65 | 50 | 0.15 | 70 | 52 |
| Clay Surf WM | T114 | 0.58 | 57 | 43 | 0.92 | 33 | 25 | 0.65 | 56 | 42 |
| | V37 | 0.20 | 74 | 62 | 0.21 | 65 | 48 | 0.12 | 80 | 62 |
| Clay Sub PCS | T114 | 0.54 | 70 | 55 | 0.93 | 38 | 30 | 0.57 | 70 | 56 |
| | V37 | 0.19 | 117 | 94 | 0.31 | 107 | 81 | 0.29 | 114 | 92 |
| Clay Sub WM | T114 | 0.51 | 73 | 58 | 0.93 | 39 | 30 | 0.61 | 68 | 53 |
| | V37 | 0.21 | 116 | 93 | 0.30 | 107 | 82 | 0.26 | 122 | 94 |

PCS—Previous Covariate Selection; WM—Wrapper Method; T: Training dataset; V: validation data set; RT: regression tree; RF: random forest; SVM: support vector machine.

The fitted model $R^2$ varied from 0.34 to 0.62 for RT models, from 0.91 to 0.95 for RF, and from 0.39 to 0.81 for SVM models. The validation RMSE, considering all 37 validation samples, varied across all sampling approach and methods of covariate selection in the ranges of 144 to 198 for the surface sand and 162 to 202 (g kg$^{-1}$) for the sand subsurface layer. For silt, the range was from 141 to 182 at the surface layer and from 89 to 102 (g kg$^{-1}$) at the subsurface layer. The RMSE range of clay was from 65 to 111 at the surface and 107 to 141 (g kg$^{-1}$) at the subsurface layer.

The RA sampling approach outperformed the TA approach for the surface and subsurface sand and silt contents, whereas surface and subsurface clay contents were best predicted using TA approach. The PCS covariate selection method was the best option to predict surface sand, and surface and subsurface silt and clay contents, whereas WM was the preferred choice only for subsurface sand prediction.

### 3.5. Relative Improvement (RI%) from Adding the Radar P-Band

Considering the combination of best results (the algorithms—RF, RT and SVM, the approach—RA or TA, and the covariate selection method—WM or PCS), the gain in accuracy of the models, with and without the P-band, was evaluated applying the RI index (%) on the $R^2$, RMSE and MAE metrics at surface and sub surface layers (Figures 7 and 8 respectively). Considering the surface layer (Figure 7), in the prediction of sand and silt, the RA approach had better results and so the metrics were separated by blocks (Urucu, Araracanga, and Jurua), how much the P-band influences the accuracy when the model generated in the

RA is transferred to other blocks was evaluated. For clay, as the TA approach performed better, the metrics do not distinguish between blocks. Note that the introduction of the P-band had a greater effect on the $R^2$ results. For the sand fraction, the introduction of the P-band allowed the $R^2$ (the proportion of the variation of a response variable is explained by the variation of other explanatory variables) to increase by 41%, 46%, and 24% for the Urucu, Araracanga, and Juruá blocks, respectively. However, when analyzing the RMSE and MAE metrics, the gain was low (<5%). In the case of silt, the introduction of P-band also increased $R^2$, but to a lesser extent (7.4%, 12%, and 10.6% for Urucu, Araracanga, and Juruá, respectively). As in the case of sand, the change in the RMSE and MAE metrics for silt prediction was low (between 0% and 1.8%). In the case of the clay attribute, the introduction of the P band did not change the metric values (RI% = 0).

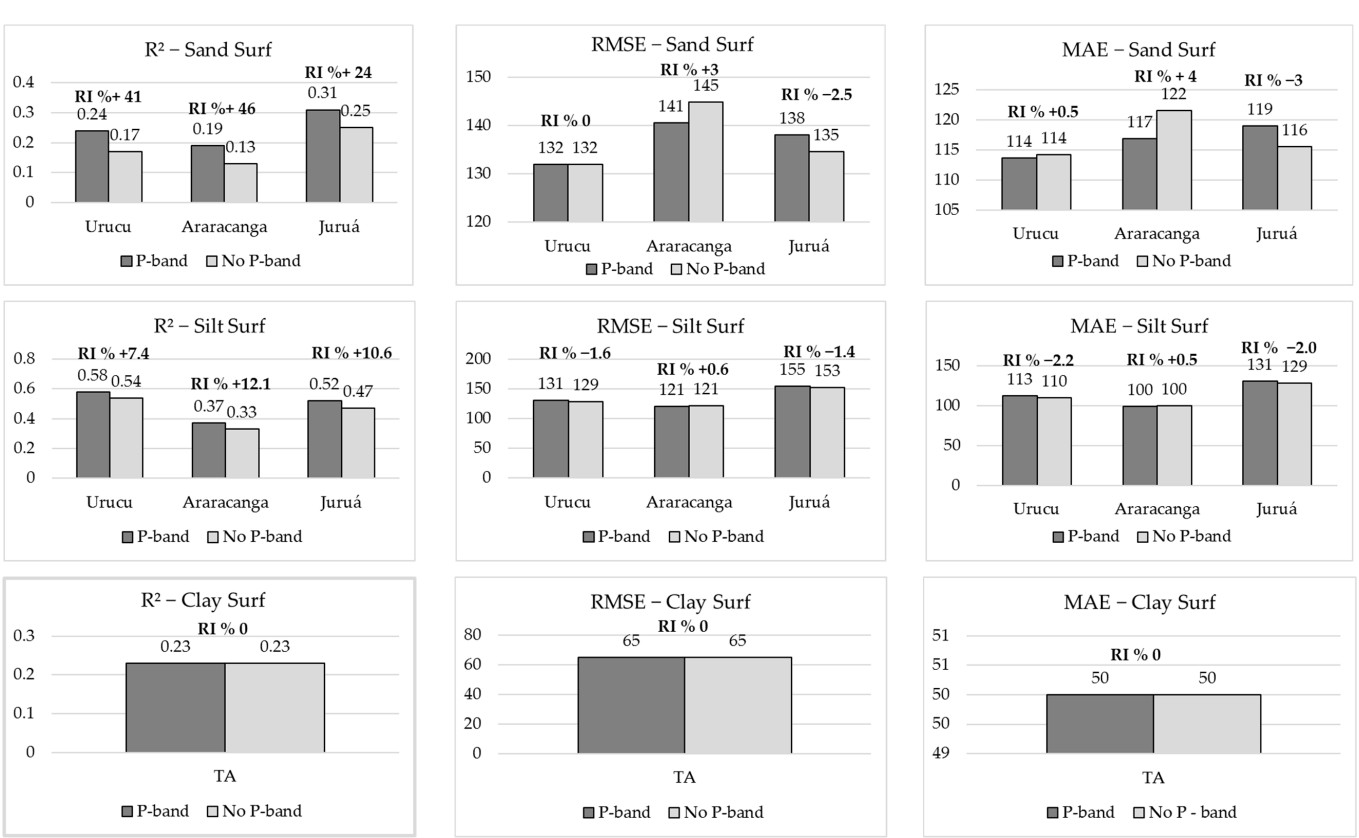

**Figure 7.** Accuracy with and without radar P-band for surface sand, silt, and clay prediction with the best model and approach.

Analyzing the subsurface layer (Figure 8), the pattern observed on the surface was maintained. In other words, the use of radar images is important to generate maps of covariates (in this case, the relief and hydrographic attributes) under native forest cover; however, the effect of the backscatter coefficient with polarization HH, by itself, did not bring a significant gain (≥10%) in the accuracy of the models (RMSE and MAE). For example, adding the P-band improved the RF predictions of clay content at the subsurface layer, with RI of the $R^2$, RMSE, and MAE of 29%, 5%, and 5%, respectively.

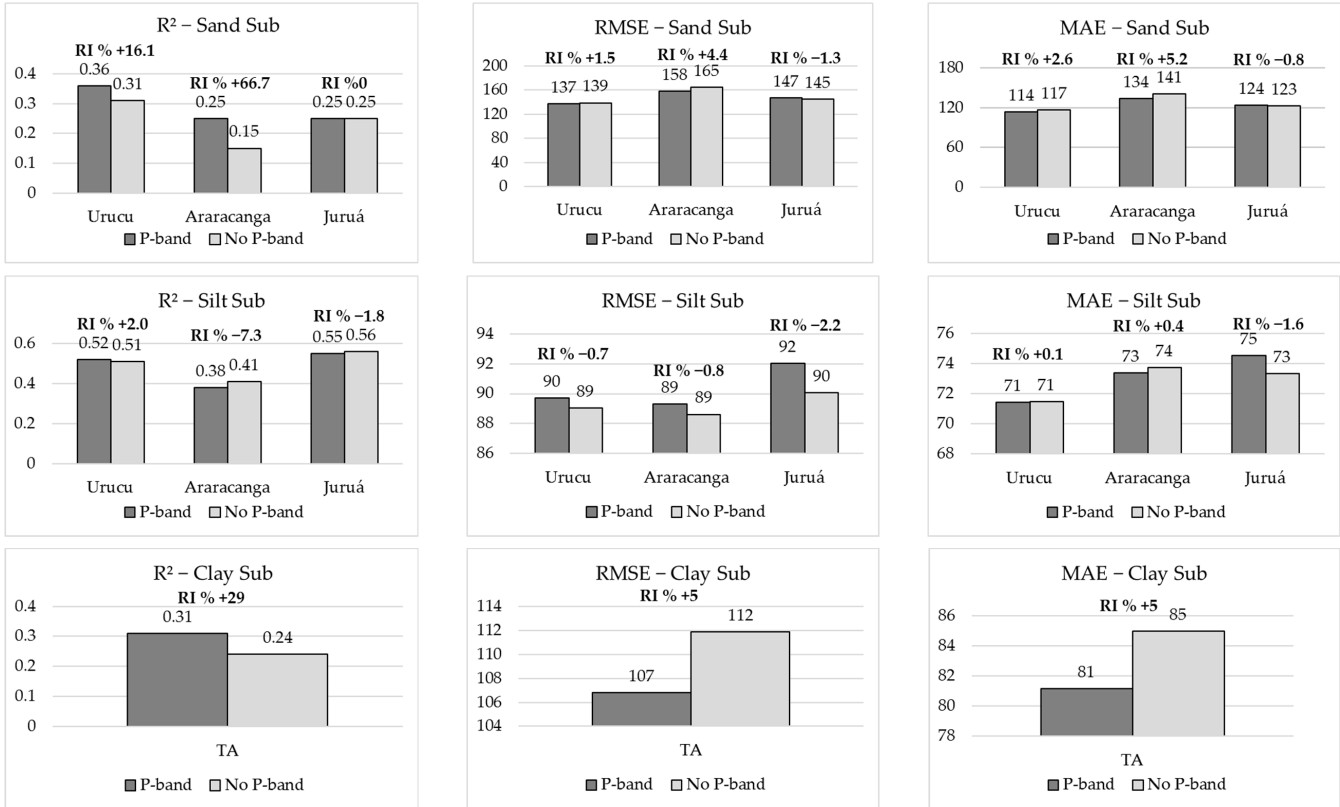

**Figure 8.** Accuracy with and without radar P-band for subsurface sand, silt, and clay prediction with the best model and approach.

Analyzing the subsurface layer (Figure 8), the pattern observed on the surface was maintained. In other words, the use of radar images is important to generate maps of covariates (in this case, the relief and hydrographic attributes) under native forest cover; however, the effect of the backscatter coefficient with polarization HH, by itself, did not bring a significant gain ($\geq$10%) in the accuracy of the models (RMSE and MAE). For example, adding the P-band improved the RF predictions of clay content at the subsurface layer, with RI of the $R^2$, RMSE, and MAE of 29%, 5%, and 5%, respectively.

### 3.6. Soil Particle Size Fraction Maps

In the study area, sand contents ranged from 303 to 721 g kg$^{-1}$ at the surface (Figure 9), and from 212 to 635 g kg$^{-1}$ at the subsurface (Figure 10), decreasing slightly with depth. The lowest sand values were predicted in hydromorphic flat tops and areas with steeper slopes (Figures 9 and 10). The highest levels of sand were present in the floodplain regions, close to the channels of the large rivers and streams, and on terraces around the main watercourse (U-shaped valleys). Large sand contents were also found in the more embedded valleys (V-shaped valleys) of slope regions. These environments are characterized by the accumulation of sandy sediments from natural erosive processes, making the lowlands clogged. In these areas, the predominant soils were classified as Aquents or Aquepts (MU2 unit).

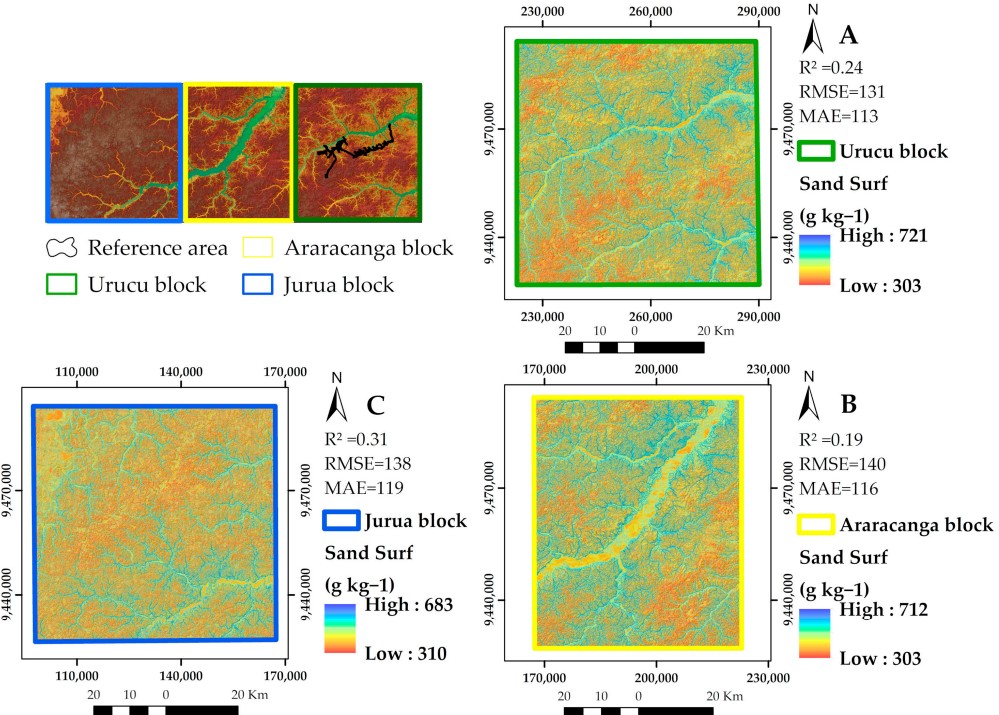

**Figure 9.** Map of the sand content at the surface layer. (Map generated using Reference Area approach, Random Forest, and Previous Covariate Selection), (**A**) Urucu block, (**B**) Araracanga block, (**C**) Jurua block.

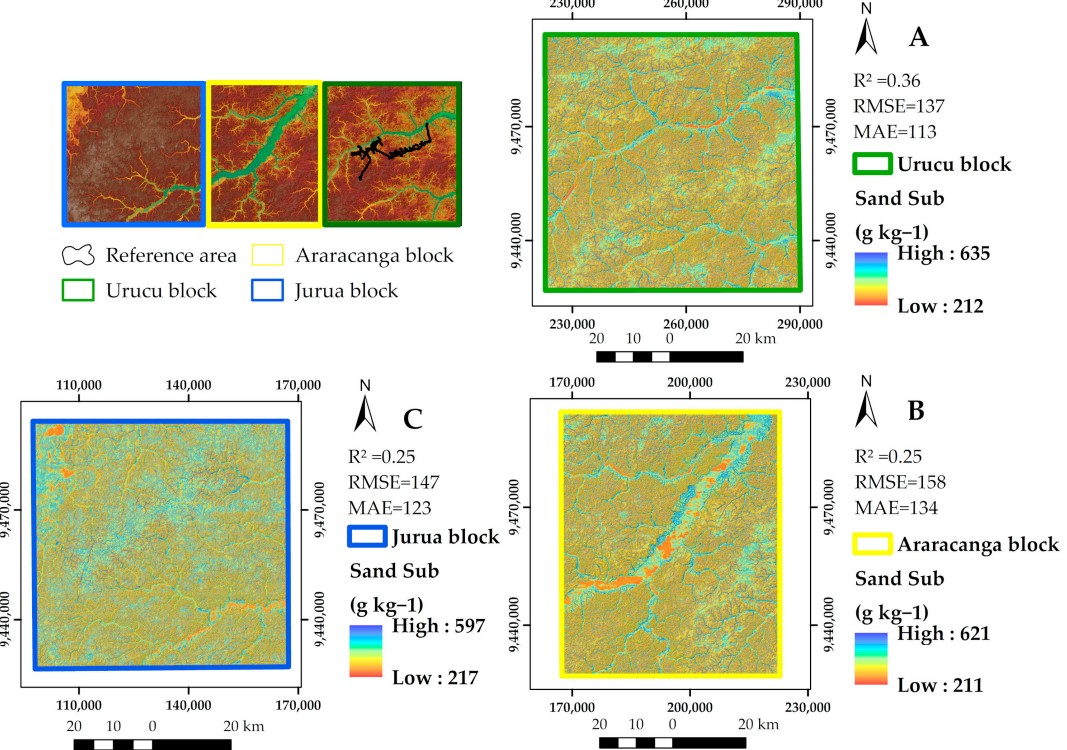

**Figure 10.** Map of the sand content at the subsurface layer. (Map generated using the Reference Area sampling design, Random Forest, and Wrapper Method), (**A**) Urucu block, (**B**) Araracanga block, (**C**) Jurua block.

Predicted silt contents varied from 209 to 577 g kg$^{-1}$ at the surface (Figure 11), and from 215 to 517 g kg$^{-1}$ at the subsurface (Figure 12). The largest silt contents were found in the areas of hydromorphic flat tops (Figures 11 and 12). These areas usually occur at the highest elevations of the study area, at the upland watershed boundaries. Flat relief and insufficient drainage characterize these areas, where there is a predominance of Hapludults, Aquults, and Aquents (MU4 unit) (Figure 6). Relevant silt values were also found in lowland regions, where Aquents (MU2) occur.

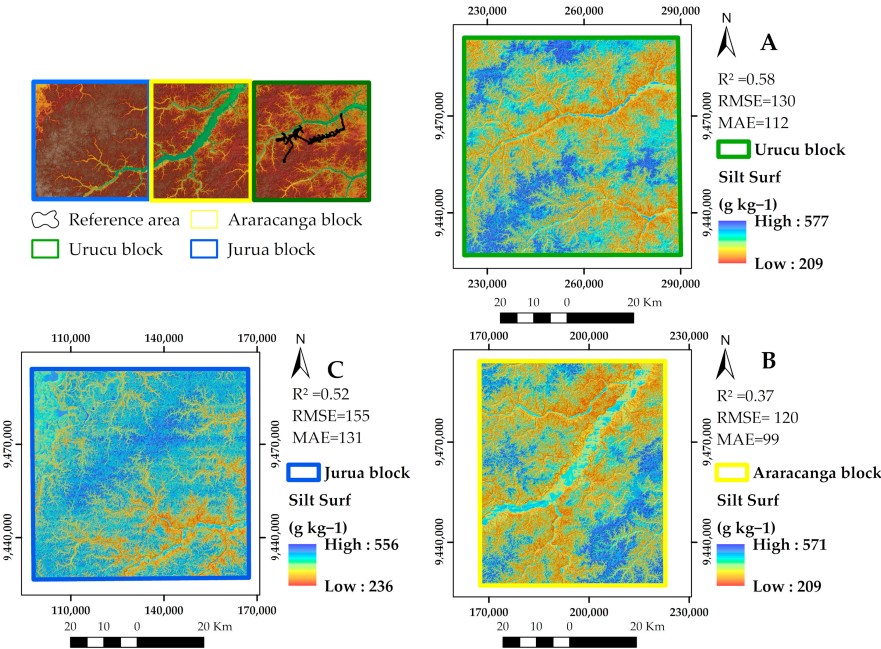

**Figure 11.** Map of the silt content at the surface layer. (Map generated using the Reference Area sampling design, Random Forest, and Previous Covariate Selection), (**A**) Urucu block, (**B**) Araracanga block, (**C**) Jurua block.

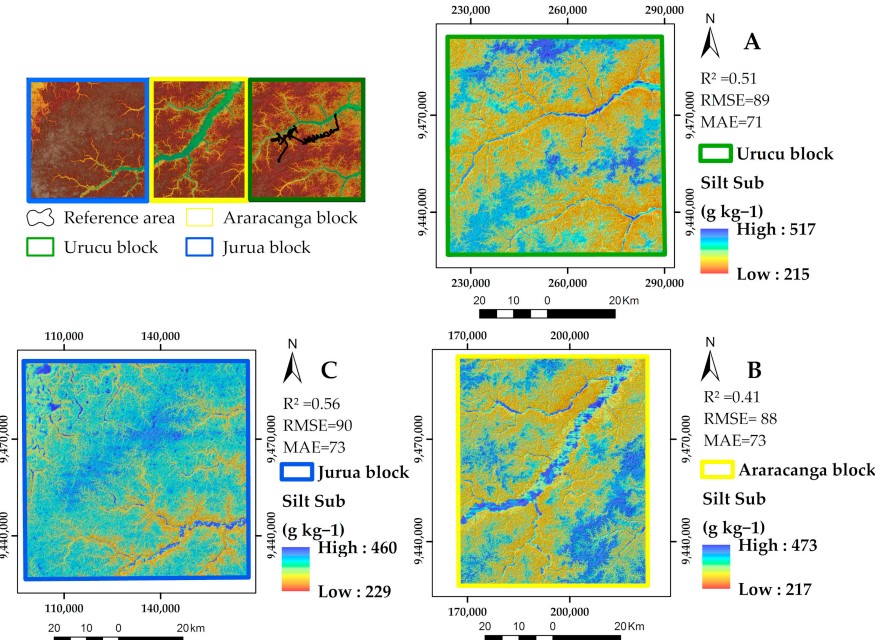

**Figure 12.** Map of the silt content at the subsurface layer. (Map generated using the Reference Area sampling design, Random Forest, and Previous Covariate Selection), (**A**) Urucu block, (**B**) Araracanga block, (**C**) Jurua block.

Predicted clay contents ranged from 47 to 303 g kg$^{-1}$ at the surface (Figure 13), and increased at the subsurface, ranging from 154 to 458 g kg$^{-1}$ at the subsurface (Figure 14). The increase of clay with depth is consistent with the occurrence of Ultisols, which present a diagnostic argillic B horizon at the subsurface. The highest clay contents occur in areas with steep slopes and well-drained tops (Figures 13 and 14). These regions were represented by the mapping units MU1 and MU3 where there is a predominance of Ultisols.

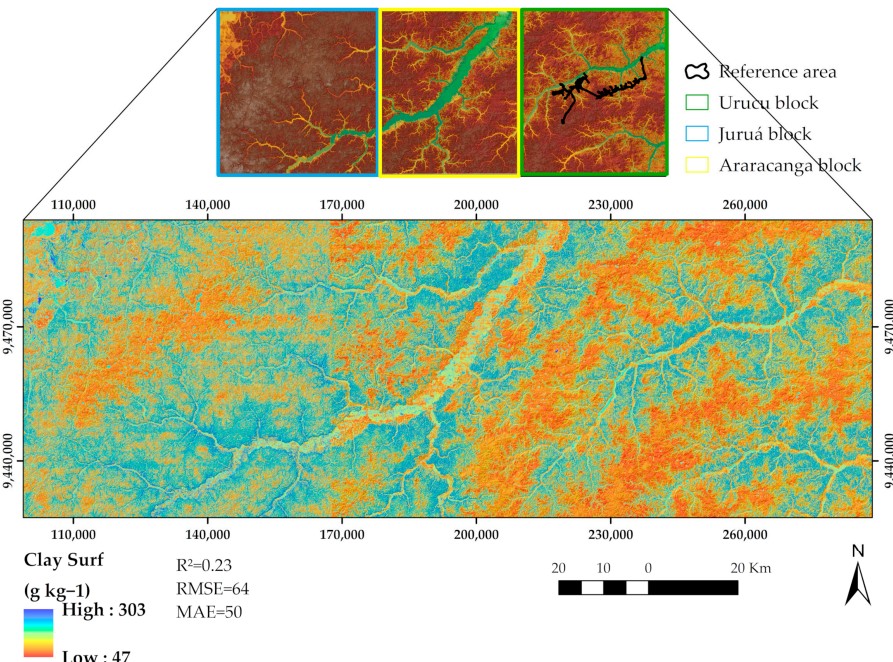

**Figure 13.** Map of the clay content at the surface layer. (Map generated using the Total Area sampling design, Random Forest, and Previous Covariate Selection).

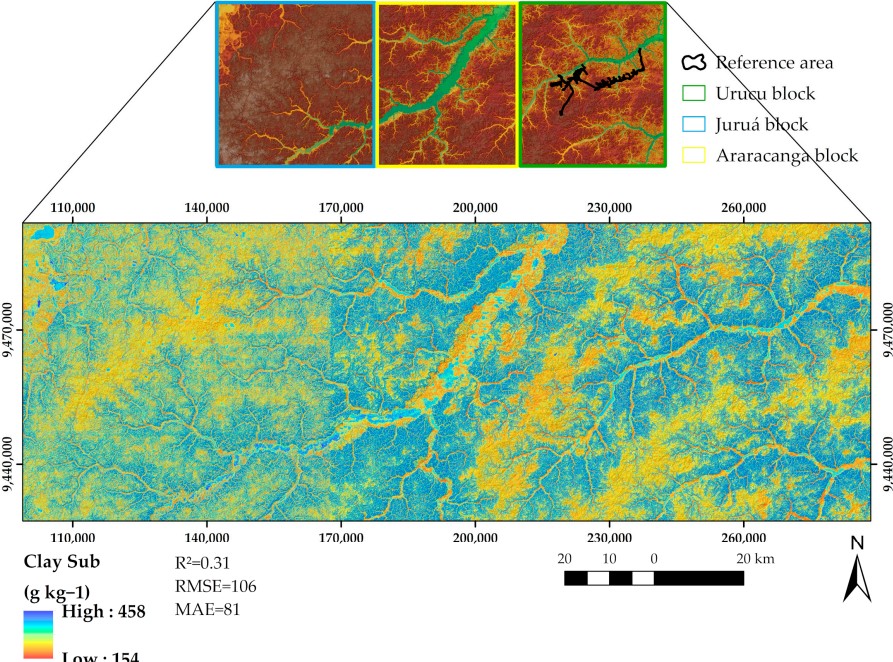

**Figure 14.** Map of the clay content at the subsurface layer. (Map generated using the Total Area sampling design, Random Forest, and Previous Covariate Selection).

## 4. Discussion

The challenge of mapping soil fraction in the Amazon rainforest comes from the difficulties in obtaining soil data that are related with the fact that a major portion of the area is covered by a dense evergreen forest, a low density of roads, with most of the territory accessed only by boat and air transport. Additionally, the difficulty of obtaining data on representative environmental covariates, because of the constant presence of clouds in the region, makes it difficult to use satellite images and aerial photos obtained by passive (optical, infrared) remote sensors. Despite all these limitations, the results of this study illustrate the potential advantages of using ML algorithms associated with remote sensor covariates (terrain attributes and P-band of airborne radar) and RA approach to map particle size fractions in this region.

The comparison of these approaches highlighted that the non-linear model introduced significant improvements in the prediction of soil texture fractions and consequently ML are potentially superior to linear methods of spatial prediction of soil texture [44]. Additionally [45], ML algorithms, in this case Support Vector Regression (SVR), produced the best prediction accuracy compared with the geostatistical interpolation techniques. The results of this study, with best the prediction for the RF model, corroborate those of ref. [46], which also used radar data to estimate soil texture and obtained better results with RF than SVM. As already highlighted by refs. [8,47], the maximum silt values are relatively high when compared with the average contents found in Brazilian soils. According to ref. [8], in the Amazon region, silt greater than 400 g kg$^{-1}$ are manly found in hydromorphic soils in the region of hydromorphic soils, which are not only found on lowlands but also in broad plateaus located in higher-altitude regions [7]. These regions have specific environmental characteristics (Figure 6) that allowed a good capture of patterns by the environmental covariates, which resulted in good prediction results for this fraction.

In general, both the correlation coefficients (Figure 4) and the most important covariates used to predict and map soil particle size fractions by RF (Figure 5) coincide with the hypotheses raised in ref. [9], as well as with previous studies in the region [7,8,48].

Some of these covariates also appear as important predictors of soil particle size fractions in ref. [49], where slope and TWI predictors had 80% of the importance for predicting surface clay (0 to 30 cm), and TWI and MRVBF were important covariates for silt prediction. In Iran, ref. [2] found TWI as one of the most important covariates for clay prediction, and similarly TWI and MRVBF were important ones for silt prediction.

The spatial patterns of the soil particle size fractions found in this study corroborate the results of ref. [8] carried out in the same study region.

A few studies have investigated the potential of P-band in mapping soil properties, most of them focus on the soil moisture and soil dielectric variations [20,22]. It is even rarer to study the P-band in the soil mapping or vegetation in the Brazilian Amazon [50]; for the authors, P-band data can make a substantial contribution to the development of models in tropical rainforest regions, especially in those areas where it is difficult to obtain data from optical sensors. Although it is not possible to compare the results with other studies, as there has been no work conducted on the use of P-band to predict soil texture, our results showed that it has great potential to improve the predictions of clay, silt, and sand fractions at the surface and subsurface, and new studies with more soil data are required to formulate better conclusions. Besides, if the VV polarization of the P-band image were available, perhaps it would be possible to extract greater knowledge of the interaction of the ratios and crosses of polarizations with granulometric fractions. For example, ref. [51], working with the X-Band, found that the sensitivity of soil texture is better observed at higher-incidence angles than lower-incidence angles in both polarizations, i.e., HH- and VV-pol. Besides, changes in soil texture are also sensitive to polarization and it was observed that VV-pol is more sensitive than HH-pol for different soil texture field. On the other hand, ref. [52], also working with the X-Band, found that a strong change in specular scattering coefficient is observed by changing the sand percentage in soil for HH polarization, while in the case of VV-polarization a lesser change is observed. It is difficult to observe the change in

specular scattering coefficient with change in soil texture when the surface is considered as rough. Finally, the authors highlighted that it is important to minimize the roughness effect while observing the texture with specular scattering and that for higher-incidence angles (P50°), the distinction in soil texture fields are clearly observable on the basis of the copolarization ratio.

The Amazon region has peculiar characteristics that demand an enormous logistical, financial, and personal effort to survey soils. It is not by chance that the major soil surveys date from the 70′s and 80′s [9] and they are exploratory or reconnaissance types. Despite all the limitations imposed by the condition of the region, this study showed that the RA approach can reduce logistical, financial, and personnel costs. In addition, the use of covariables such as P-band, which is able to surpass the tree canopy and suffers little or no interference from clouds, combined with covariate selection methods and the training of robust ML algorithms can greatly increase the prediction results, producing more detailed and very useful maps.

## 5. Conclusions

This work investigated the use of remote sensing covariates derived from airborne synthetic aperture interferometric radar images to predict soil surface and subsurface sand, silt, and clay contents in the Brazilian Central Amazon. A Reference Area sampling design was proposed to reduce costs and expedite soil survey was contrasted against a random sampling design (that is, Total Area sampling), and combined with three machine learning methods (RT, RF, and SVM) and two covariate selection approaches (WM and PCS).

The RA approach was the best sampling option, deriving the least errors, for surface and subsurface silt and sand content prediction. Total Area random sampling was preferred for surface and subsurface clay content prediction, though the errors were similar to those from the RA approach. The RA was 80 km$^2$, whereas the whole area to be mapped was 13.440 km$^2$. This means that a tiny fraction of 0.6% of the total area served to collect soil and remotely sensed relief and P-band data to train soil particle size prediction methods, and transfer them to the whole area, composed by three relatively huge exploration blocks. Thus, the RA approach combined with remote sensing is recommended for expediting soil mapping and saving costs, especially in large areas.

From the relief attributes derived from the DEM, it was possible to establish relationships between the soil particle size fractions and the landscape. The selection of covariates (PCS) obtained, in general, better results than the all-in WM option that is commonly employed in digital soil mapping studies. The most important covariates to predict the soil particle size fractions in the Central Amazon region were CI, LF, MRRTF, MRVBF, TWI, slope, and ProfC for all fractions, in addition to the radar P-band backscatter coefficient for surface sand and clay contents.

Random forest outperformed RT and SVM for all soil particle size fractions and both layers. It is recommended for its robustness and ease to implement in free and open-source software. The P-band backscatter coefficient was considered an important covariate for the prediction of surface sand and clay contents by RF, showing its potential use for mapping these attributes.

**Author Contributions:** Conceptualization: M.B.C. and A.C.d.S.F.; methodology: M.B.C. and A.C.d.S.F.; software: A.C.d.S.F. and E.M.C.; validation: M.B.C.; formal analysis: M.B.C. and A.C.d.S.F.; investigation: M.B.C., A.C.d.S.F., É.F.M.P. and G.M.V.; resources: M.B.C.; data curation: M.B.C., A.C.d.S.F. and E.M.C.; writing—original draft preparation: A.C.d.S.F.; writing—review and editing: M.B.C., A.C.d.S.F., E.M.C., É.F.M.P., M.M.d.N. and G.M.V.; visualization: M.B.C., A.C.d.S.F. and E.M.C.; supervision: M.B.C.; project administration: M.B.C.; funding acquisition: M.B.C. All authors reviewed the manuscript. All authors have read and agreed to the published version of the manuscript.

**Funding:** This research received no external funding.

**Acknowledgments:** The authors acknowledge the National Petroleum Agency (ANP) for funding the project "Digital Mapping of Soils in Oil and Gas Exploration and Production Areas—Case Studies of the North and Northeast Brazilian Fields" under the agreement number 5850.0105881.17.9 (PETROBRAS/FAPUR/UFRRJ) and Evaluation of Graduate Education (CAPES, finance code 001).

**Conflicts of Interest:** The authors declare no conflict of interest.

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
