# Peer review of "Use of Airborne Radar Images and Machine Learning Algorithms to Map Soil Clay, Silt, and Sand Contents in Remote Areas under the Amazon Rainforest"

_remotesensing, doi:10.3390/rs14225711_

Round 1
Reviewer 1 Report
After carefully reviewing the figure material and contents of these manuscripts, I decided to reject the manuscript for the following reasons: (1) The whole manuscript seems to be of low quality, including the quality of all figures and the quality of the maintext; (2) There are major defects in the work design of this study, and the description of work methods and processes is not clear; (3) The evidence presented by the author does not support the conclusion very well. For example, the author mentioned in the title that it is based on airborne radar, but the content actually deviates from the subject; (4) The method in this word lacks the support of professional knowledge and is only an application report of computer graphics and artificial intelligence classification. It is suggested to add more professional interpretations; (5) The discussion part is not good, more content needs to be added. Therefore, my decision would be rejection.Author Response
We would like to thank the reviewer for evaluating our manuscript. We also sincerely apologize for submitting the figures and tables separately from the manuscript text and also for compressing the quality of the images.

Reviewer 2 Report
I reviewed the manuscript “Use of airborne radar images and machine learning algorithms to map soil clay, silt and sand contents in remote areas under Amazon rainforest” written by Ferreira et al. I commend the authors for their work on mapping soil textures in the Amazon region using machine learning algorithms with the radar P-band. The authors did a interesting and meaningful research. Nevertheless, some questions should be considered before publication:
1. Although it is difficult to obtain soil samples in the Amazon region, the total number of 151 samples seems to be small. Not to mention that the sample has been divided into many blocks.
2. In the discussion part, the authors observed that soil moisture has a significant impact on the specular scattering coefficient. Have the authors considered the influence of soil water content on developing the model? Moreover, I guess the soil organic matter content should be high. Did the authors consider this?
3. From the website, I can only download the manuscript text without the tables and figures. I cannot intuitively find the information of input data (such as the locations, soil sample depth, etc.) and the results. I suggest the authors add the tables and figures into the manuscript.
4. To evaluate the accuracy of models, except for R2, MAE and RMSE, I think coefficient of variance (CV) should be considered for comparing the results between RA and TA.
Reviewer 3 Report
version of the manuscript does not contain any figures or tables.
I think it necessarily needs to be revised.
Reviewer 4 Report
Dear authors,
Thank you for providing an excellent manuscript.
I noted that in the introduction section there are a few missing references (e.g. Line 46-47, 49-51, 59-61, 68-70 etc.). Could you kindly ensure that missing references are included?
In the results section it would be helpful if a table with summary statistics or maps/graphs would be included. Please include this information. The discussion in the results section has to be moved to the discussion section. Missing references will have to be added. The discussion section of the manuscript is too short and should add some additional comparison.
Why is there literature cited in the results section? I recommend moving this to the discussion section.
Kind regards
Round 2
Reviewer 1 Report
Although the authors have responded to the comments and made some modifications, it has not fully responded to the five previous major issues, and the modifications are far from meeting the requirements. To sum up, I still suggest rejecting the manuscript.
Reviewer 3 Report
it's ok